# Interfacial oxygen vacancies yielding long-lived holes in hematite mesocrystal-based photoanodes

Zhujun Zhang [1], Izuru Karimata[1], Hiroki Nagashima [2], Shunsuke Muto [3], Koji Ohara [4], Kunihisa Sugimoto [1,4,5] & Takashi Tachikawa [1,2]*

Hematite ($\alpha$-$Fe_2O_3$) is one of the most promising candidates as a photoanode materials for solar water splitting. Owing to the difficulty in suppressing the significant charge recombination, however, the photoelectrochemical (PEC) conversion efficiency of hematite is still far below the theoretical limit. Here we report thick hematite films ($\sim$1500 nm) constructed by highly ordered and intimately attached hematite mesocrystals (MCs) for highly efficient PEC water oxidation. Due to the formation of abundant interfacial oxygen vacancies yielding a high carrier density of $\sim$10$^{20}$ cm$^{-3}$ and the resulting extremely large proportion of depletion regions with short depletion widths (<10 nm) in hierarchical structures, charge separation and collection efficiencies could be markedly improved. Moreover, it was found that long-lived charges are generated via excitation by shorter wavelength light (below $\sim$500 nm), thus enabling long-range hole transfer through the MC network to drive high efficiency of light-to-energy conversion under back illumination.

[1] Department of Chemistry, Graduate School of Science, Kobe University, 1-1 Rokkodai-cho, Nada-ku, Kobe 657-8501, Japan. [2] Molecular Photoscience Research Center, Kobe University, 1-1 Rokkodai-cho, Nada-ku, Kobe 657-8501, Japan. [3] Electron Nanoscopy Section, Advanced Measurement Technology Center, Institute of Materials and Systems for Sustainability, Nagoya University, Furo-cho, Chikusa-ku, Nagoya 464-8603, Japan. [4] Diffraction and Scattering Division, Center for Synchrotron Radiation, Japan Synchrotron Radiation Research Institute, 1-1-1 Kouto, Sayo-cho, Sayo-gun, Hyogo 679-5198, Japan. [5] Institute for Integrated Cell-Material Sciences (iCeMS), Kyoto University, Yoshida-Ushinomiya-cho, Sakyo-ku, Kyoto 606-8501, Japan. *email: tachikawa@port.kobe-u.ac.jp

Solar water splitting ($H_2O + h\nu \rightarrow 1/2\ O_2 + 2\ H^+$), a promising way to turn solar light into clean energy, has inspired significant research but remains a great challenge in improving efficiency and reducing cost for practical applications[1–6]. With the superiority of higher efficiency and stability than molecular systems and lower cost than simple combination of photovoltaics and electrolysis, photoelectrochemical (PEC) water-splitting system is particularly desirable[3,4]. Owing to the advantages of abundance, long-term stability in alkaline solution, and a bandgap (~2.1 eV) suitable for absorption of visible light, hematite ($\alpha$-$Fe_2O_3$) has been regarded as an ideal photoanode semiconductor for PEC water splitting[7]. However, the short lifetime (picosecond time scale) and diffusion length (2–4 nm) of photogenerated charges due to the significant charge recombination in the bulk or at the surface lower the PEC performance of hematite photoanodes[8–10]. Therefore, most of the efforts have been focused on the fabrication of nanostructured thin films[11–13] and the passivation of surface defects[7] to suppress the recombination. However, the conflict between charge separation and light harvesting ($\alpha^{-1} = 0.12\ \mu m$ at 550 nm for hematite, where $\alpha$ is absorption coefficient) has significantly limited their application[7,14]. This is why state-of-the-art hematite photoanodes[15–18] still showed poor efficiency as compared with the theoretical solar-to-hydrogen (STH) efficiency of ~15% (or photocurrent generation of ~13 mA cm$^{-2}$ at 1.23 V vs. RHE under 1-sun illumination)[7]. In addition, the requirement of front illumination (irradiate the hematite film surface through electrolyte) to drive the water oxidation also largely limited the cell design due to light scattering or shadowing by evolved gas bubbles. However, thick hematite films enabling high efficiency of water oxidation via back illumination have not been reported so far.

A few examples of mesoporous hematite structures composed of aggregated nanostructures have been demonstrated to be effective for PEC water splitting[16,19]. However, owing to the disordered structures where nanoparticles collide and aggregate randomly, it is still hard to identify the champion nanostructures in a single electrode which may contribute most of the electrode's photocurrent[16]. In addition, the crystal–crystal interfaces or grain boundaries may act as recombination sites due to the lattice mismatch in crystals with different orientation and thus limit charge transport[20]. Mesocrystals (MCs) are superstructures of nanoparticles with a specific preferable mutual orientation[21,22]. Tachikawa et al. demonstrated that MCs of metal oxides, such as $TiO_2$ and $SrTiO_3$, have superior efficiencies in charge separation and transport between primary nanocrystals compared with conventional nanocrystal systems owing to their highly ordered structures[23–27]. Besides, the interfacial atomic structures between facing nanocrystal subunits inside the MC might be partly adjusted to reduce the recombination sites, possibly resulting in the improvement of intergranular electronic conductivity. Therefore, the hieratical assembly of MCs with minimum disorders and appropriate adjustment of the interface may further improve the PEC performance.

Herein, we demonstrate impressive performance of hematite MC-based photoanodes with thickness of ~1500 nm constructed by ~15 layers of highly oriented and intimately attached Ti-modified hematite MCs (Ti–$Fe_2O_3$ MCs). The photocurrent density at 1.23 V vs. reversible hydrogen electrode (RHE) (Supplementary Note) reaches to 2.5 mA cm$^{-2}$ with an onset potential of 0.8 V vs. RHE. Further analytical results illustrate that abundant interfacial oxygen vacancies ($V_O$) formed during partial sintering at the interfaces between the well-oriented nanocrystals within the MC can largely increase the carrier density for efficient charge separation and transport. In addition, the unique mesoporous structures with extremely high proportion of thin depletion regions could greatly improve the charge-collection efficiency. Moreover, excitation with short-wavelength light generates long-lived charges, resulting in a considerable number of holes (~14%) diffused far from the excited region to the unexcited one along the intimately connected MCs. These characteristics of hematite MC-based photoanodes with rich interfacial $V_O$ enable high PEC performance and open a useful strategy to design the superstructure-based systems for efficient solar fuel production.

## Results

**Synthesis and characterization of the hematite MCs.** The highly stable Ti–$Fe_2O_3$ MCs were synthesized via an additive-free solvothermal self-assembling method using N,N-dimethylformamide (DMF) and methanol as the mixed solvent, and Fe$(NO_3)_3$·9$H_2O$ and $TiF_4$ as metal precursors (Supplementary Fig. 1). The representative Ti–$Fe_2O_3$ MCs obtained with a $TiF_4$/Fe$(NO_3)$·9$H_2O$ molar ratio of 1:10 show a uniform discus-like structure as revealed by transmission microscopy (TEM) (Fig. 1a; Supplementary Fig. 2) and high-angle annular dark-field scanning TEM (HAADF-STEM) images (Supplementary Fig. 3a). Ti–$Fe_2O_3$ MCs with diameters of ~300 nm and thickness of ~90 nm are composed of closely stacked nanocrystal subunits (~30 nm). The corresponding selective area electron diffraction (SAED) pattern captured over the whole region of MC exhibits a single-crystal-like spot pattern (Fig. 1a, inset), indicating that the nanocrystal subunits inside the MC are highly ordered and crystallographically aligned, which can be attributed to particle-to-particle interaction[28]. The SAED pattern displays well-defined diffraction spots of the (−210), (−120), and (110) planes along the [001] zone axis, which possess the central angles of 60º. The high-resolution TEM (HRTEM) image obtained from the region comprised two adjacent nanocrystals with an interface clearly indicates parallel crystal lattices (Fig. 1b), further proving the structural characteristics of MC[29]. There are three sets of lattice fringes with lattice spacing (d) of 0.25 nm, which corresponds to the (110) plane of hematite, and with interfacial angle of 60º, which is consistent with the SAED result. The elemental mapping images of the typical Ti–$Fe_2O_3$ MC in Supplementary Fig. 3a visualize the similar spatial distributions of Fe, O, and Ti elements on the surface. The content of Ti determined from energy dispersive X-ray spectrometry (EDX) spectrum is ~8.5%, which is close to the concentration of the $TiF_4$ precursor added (Supplementary Fig. 3b). The mesoporous structure and surface areas were confirmed by $N_2$ sorption isotherms and the corresponding pore-size distribution curve (Supplementary Fig. 4). The TEM images and corresponding SAED pattern obtained from the whole MC particle in Fig. 1c indicate that Ti–$Fe_2O_3$ MCs maintain their size, morphology, and crystal structure even after high-temperature annealing at 700 °C (Fig. 1c; Supplementary Fig. 5a, b). Mesopores with diameters ranging from 8 to 40 nm formed inside the MC after the thermal treatment as seen from the HAADF-STEM image in Supplementary Fig. 6.

HAADF-STEM with electron energy loss spectroscopy (EELS) measurements can give useful information about the local composition, structure, and chemical states of the samples[30–32]. Figure 1d shows the HAADF-STEM image of a typical Ti–$Fe_2O_3$ MC particle after being annealed at 700 °C in air. The corresponding EELS composition map is given in panel e. It is obvious that most of the Ti species are segregated to the outer surface and edges of the pores. To further analyze the distribution of the elements in the crystal, EEL spectra were acquired in different regions (Fig. 1d). As shown in Fig. 1f, only Ti-$L_{2,3}$ and O-K signals were detected on the edge of the MC particle (Region 1 in Fig. 1d), suggesting the presence of $TiO_2$ overlayer. The thicknesses of $TiO_2$ layers are estimated from <1 to ~4 nm

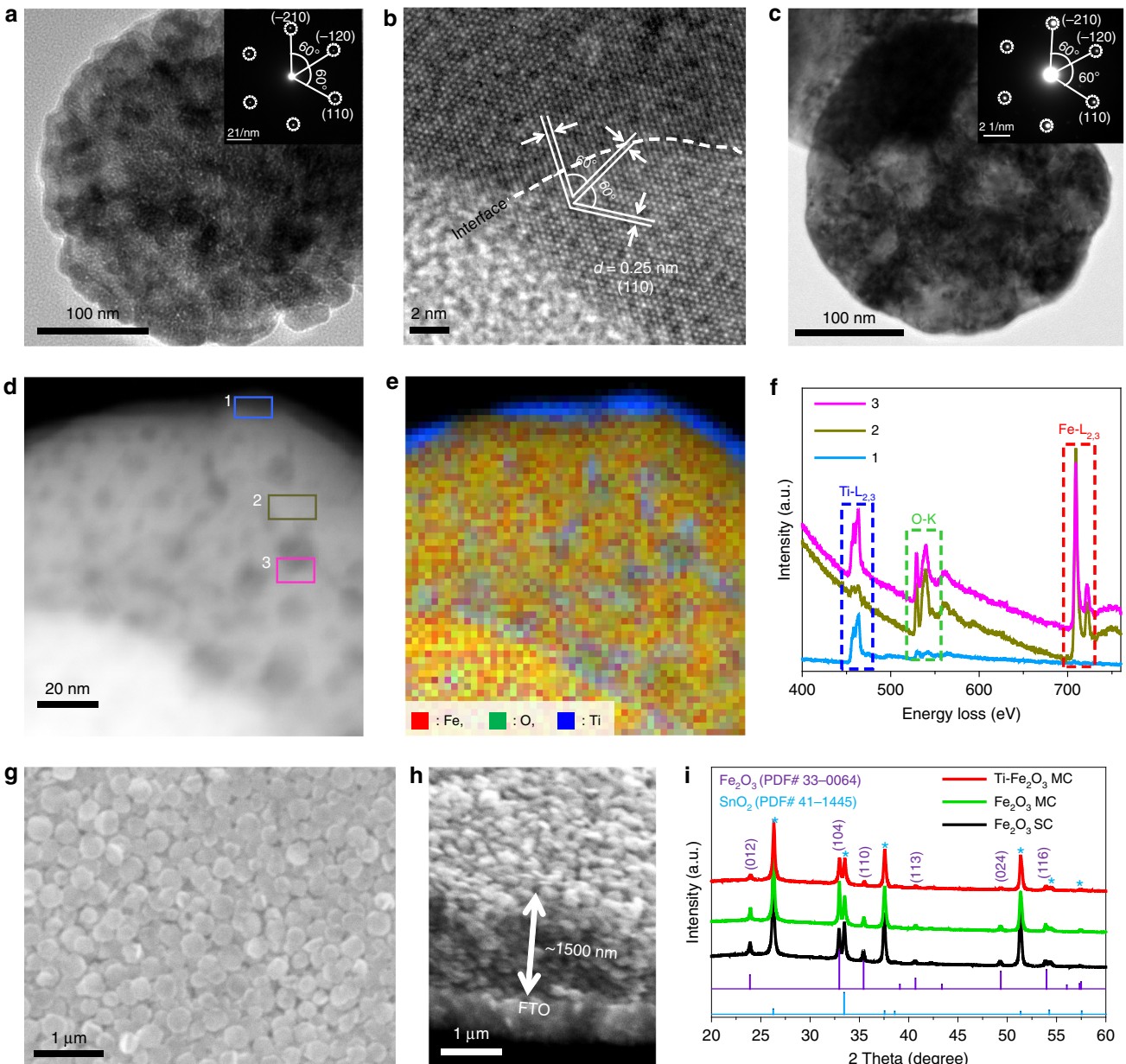

**Fig. 1** Morphological characterization of Ti–Fe$_2$O$_3$ MC. **a** TEM and corresponding SAED (inset), and **b** HRTEM image of the as-synthesized Ti–Fe$_2$O$_3$ MC. **c** TEM and corresponding SAED (inset), **d** HAADF-STEM image and **e** corresponding EELS chemical composition maps of a typical Ti–Fe$_2$O$_3$ MC particle collected after annealing at 700 °C. **f** EEL spectra of the selected regions in panel **d**. **g** Top-view, and **h** cross-sectional SEM images of the optimized Ti–Fe$_2$O$_3$ MC photoanode. **i** XRD patterns of the Ti–Fe$_2$O$_3$ MC, Fe$_2$O$_3$ MC, and Fe$_2$O$_3$ SC photoanodes

(Fig. 1e; Supplementary Fig. 7). It is noteworthy that TiO$_2$ (most probably rutile phase) grew on the surface of hematite as suggested by the high-resolution HAADF-STEM images, the corresponding fast Fourier transform (FFT) patterns, inverse FFT images (Supplementary Fig. 7), and XRD peak attributed to rutile (110) (Supplementary Fig. 8). On the smooth surface (Region 2 in Fig. 1d), Ti-L$_{2,3}$ signals are very weak, implying that a small portion of Ti$^{4+}$ ions were doped into the crystal. A higher concentration of Ti$^{4+}$ ions were observed near the edges of the pores (Region 3 in Fig. 1d), again indicating the formation of thin TiO$_2$ layers. This conclusion was further confirmed by X-ray photoelectron spectroscopy (XPS) and synchrotron-based X-ray total scattering measurements (Supplementary Figs. 9–11).

The morphology and thickness of the prepared photoanodes were characterized by scanning electron microscopy (SEM). As shown in Fig. 1g, Ti–Fe$_2$O$_3$ MC photoanode is constructed by

highly oriented and intimately attached Ti–Fe$_2$O$_3$ MCs. The optimized film with thickness of ~1500 nm consists of ~15 layers of uniform Ti–Fe$_2$O$_3$ MCs according to the cross-sectional SEM image (Fig. 1h; Supplementary Fig. 12). For comparison, unmodified hematite MC (Fe$_2$O$_3$ MC) and Fe$_2$O$_3$ single-crystal (SC) photoanodes with a similar film thickness were prepared (Supplementary Figs. 13, 14). The crystal orientation of the particles on fluorine-doped tin oxide (FTO) glasses was analyzed by X-ray diffraction (XRD) (Fig. 1i). The observed diffraction peaks were identified according to the standard PDF card (PDF#33–0664), and no other crystal phases were detected by XRD under the typical conditions, except the cassiterite phase of SnO$_2$ (PDF#41–1445) in the FTO substrate as marked by the asterisks. It should be noted that the higher ratios of (104)/(110) indicate the existence of predominantly (104) oriented hematite for all the photoanodes. The grain sizes in Ti–Fe$_2$O$_3$ and Fe$_2$O$_3$

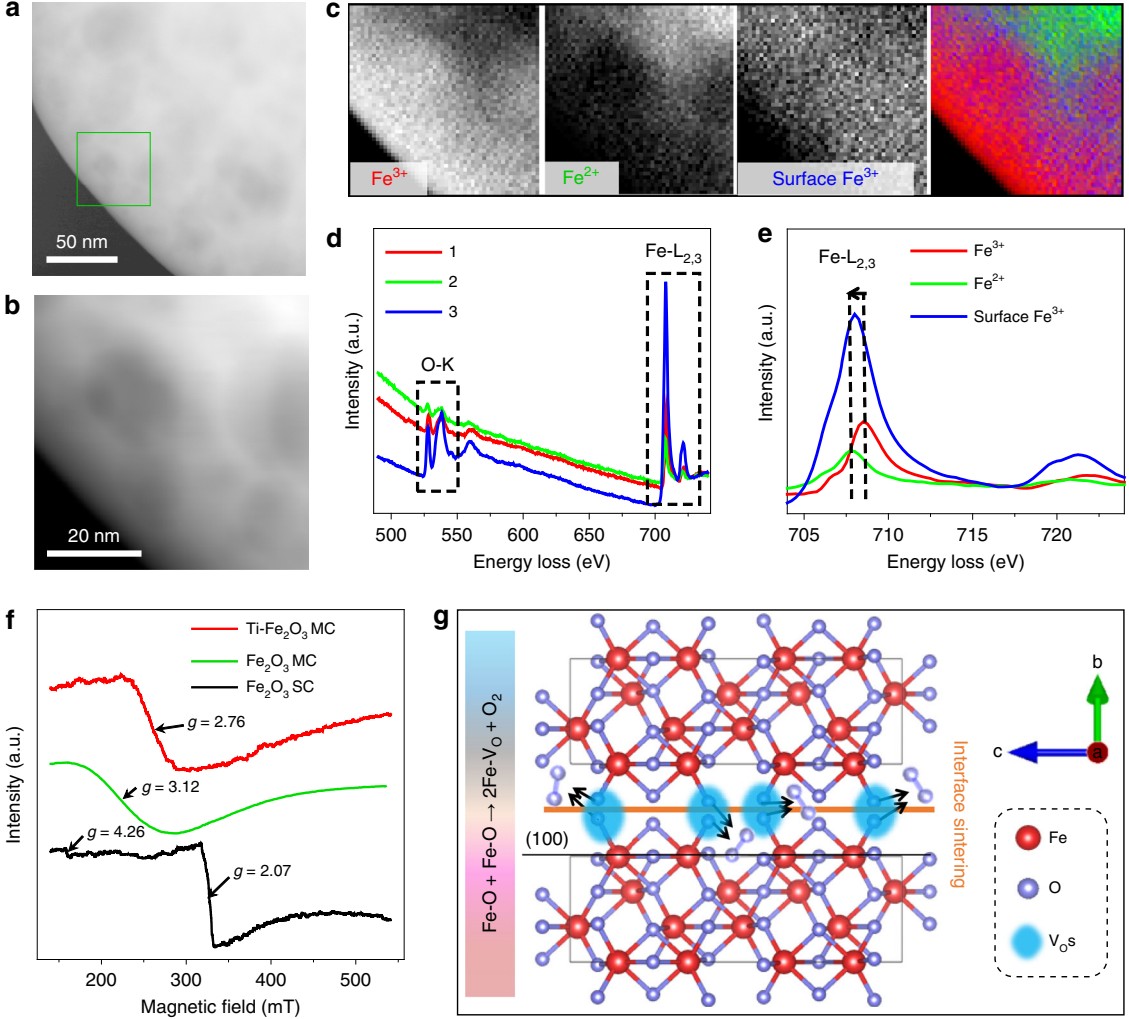

**Fig. 2** Interfacial oxygen vacancies ($V_O$) in MCs. **a**, **b** HAADF-STEM images of $Fe_2O_3$ MC and **c** corresponding EELS chemical composition maps of panel **b**. **d** EEL spectra of panel **b** and **e** corresponding EELS Fe-$L_{2,3}$ spectra of the selected region in panel **b**. The rightmost one of panel **c** is merged image of components 1 (red), 2 (green), and 3 (blue). **f** EPR spectra of the samples. **g** Schematic illustration of the $V_O$ formation in MC, where (100) facets are tentatively assumed to form the interface

MCs calculated using Scherrer equation are ~30 nm, which is close to the sizes of the nanocrystal subunits in as-synthesized MCs verified from TEM images (Supplementary Figs. 2, 13).

**Abundant interfacial oxygen vacancies ($V_O$) in MCs**. The properties of semiconductors are closely related to intrinsic lattice defects, as well as their structure and composition. To analyze the defects in MCs, STEM-EELS analysis was carried out. Figure 2a, b is the HAADF-STEM images of a typical $Fe_2O_3$ MC. As demonstrated in Fig. 2c, d, three components (1, 2, and 3) are clearly separated by multivariate analysis[33] of the spectrum image data obtained in the area of Fig. 2b. The components 1 and 2 can be assigned to oxides containing $Fe^{3+}$ and $Fe^{2+}$, respectively, form characteristic Fe-$L_{2,3}$ spectra (Fig. 2e). These components are considered to be mostly located inside the MC from their high background intensity (Fig. 2d). Importantly, $Fe^{2+}$ is distributed inside rather than the edge and pores without apparent spatial overlap with $Fe^{3+}$ (Fig. 2c). These features reflect the grain boundaries in the depth direction, although the interfaces between nanoparticles inside the MC itself almost disappeared due to sintering. In other words, $V_O$ are likely to form in the regions where fusion of neighbored nanoparticles occurred

during sintering. The component 3 probably originates from surface $Fe^{3+}$ by taking into consideration its low background intensity and the $L_3$ peak slightly shifted to the lower energy side compared with those of the standard $Fe^{3+}$ (Fig. 2e). Qualitatively similar results are obtained for different MCs (Supplementary Fig. 15). The valence states of iron in Ti–$Fe_2O_3$ MC were also identified (Supplementary Figs. 16, 17).

To further confirm the presence of abundant $V_O$ in the MC samples, electron paramagnetic resonance (EPR) measurements were carried out at 6 K (Fig. 2f). The two main signals of the $Fe_2O_3$ SC sample ($g = 2.07$ and 4.26) were assigned to $Fe^{3+}$ ions coupled via exchange interactions and $Fe^{3+}$ ions in rhombic and axial symmetry sites, respectively[34,35]. However, only single broad resonance lines centered at $g = 3.12$ and 2.76 were observed for the $Fe_2O_3$ MC and Ti–$Fe_2O_3$ MC samples, respectively. The change of the lineshape indicates a ferromagnetic resonance due to the interaction between $Fe^{3+}$ and $Fe^{2+}$ (i.e., $V_O$)[34,35]. Thus, these results convincingly reveal that the MC samples possess more $V_O$ than the SC sample. For comparison, $V_O$ was not detected for Ti–$Fe_2O_3$ MCs and $Fe_2O_3$ MCs without high temperature annealing (Supplementary Fig. 18). The formation of $V_O$ was also suggested by the O 1 s XPS spectra, XPS depth profiles, and steady-state UV–visible diffuse reflectance spectra (Supplementary

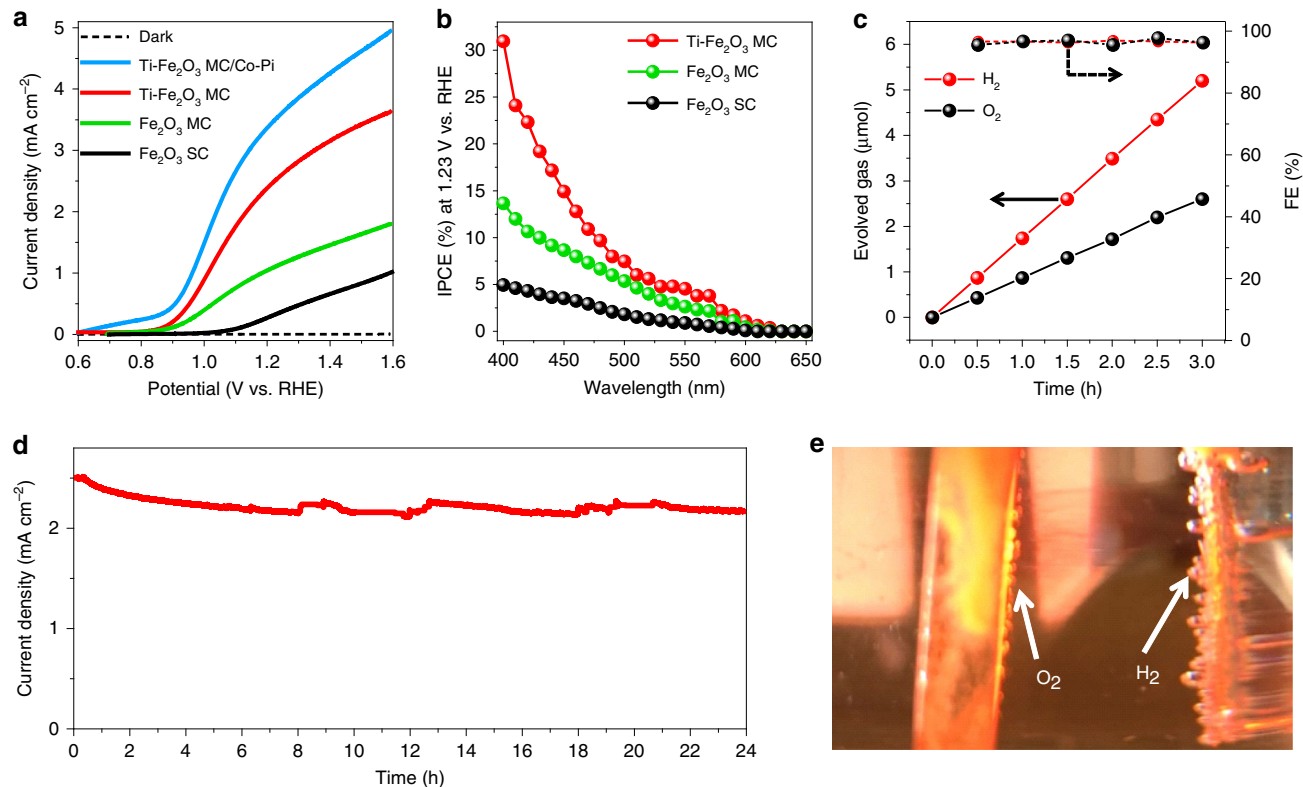

**Fig. 3** PEC performance. **a** Current density–voltage curves of Ti–Fe$_2$O$_3$ MC/Co-Pi, Ti–Fe$_2$O$_3$ MC, Fe$_2$O$_3$ MC, and Fe$_2$O$_3$ SC photoanodes prepared by 50 cycles of spin-coatings measured under back illumination with AM 1.5 G simulated sunlight in 1.0 M NaOH. **b** IPCE curves measured at 1.23 V vs. RHE for the Ti–Fe$_2$O$_3$ MC, Fe$_2$O$_3$ MC, and Fe$_2$O$_3$ SC photoanodes. **c** Gas evolved from Ti–Fe$_2$O$_3$ MC photoanode and Pt counter electrode with an applied potential of 1.23 V vs. RHE under back illumination during 3 h and the corresponding Faradaic efficiencies (FEs). **d** Current density–time curve of the Ti–Fe$_2$O$_3$ MC photoanode measured at 1.23 V vs RHE under back illumination. **e** Photograph showing gas evolution from Ti–Fe$_2$O$_3$ MC photoanode and Pt counter electrode under an applied potential of 1.23 V vs. RHE

Figs. 19, 20, and 21, respectively). Based on the above results, we propose that the partial sintering of highly ordered interfaces between the nanocrystal subunits inside the MC at high temperature could leave small interior spaces with oxygen deficiency (i.e., oxygen-deficient interfaces), and thus oxygen atoms in the hematite lattice (Fe–O) easily escape to create V$_O$ (Fe–O + Fe–O → 2Fe–V$_O$ + O$_2$) (Fig. 2g) owing to the lower formation energy of V$_O$ in metal oxides compared with cation interstitials[36]. In contrast, the Fe$_2$O$_3$ SC sample without such ordered interfaces between nanocrystals would hardly generate V$_O$ when being annealed under the same conditions.

**PEC performance of MC photoanodes**. The PEC performance of Ti–Fe$_2$O$_3$ MC photoanodes fabricated with different Ti precursor concentrations is provided in Supplementary Fig. 22. The photocurrent density of the Ti–Fe$_2$O$_3$ MC photoanode reaches a maximum at a Fe:Ti molar ratio of 1:0.1, and higher Ti/Fe ratios lead to a decrease in the photocurrent density.

The PEC performance of the optimized Ti–Fe$_2$O$_3$ MC photoanode is displayed in Fig. 3 along with the results of reference Fe$_2$O$_3$ MC and SC photoanodes. Figure 3a shows the current density–voltage curves measured in the dark and under back illumination with simulated sunlight (AM 1.5 G, 100 mW cm$^{-2}$) in 1.0 M NaOH (pH 13.6). The dark currents in the voltage range applied here are negligible for all samples. The photocurrent density obtained for the Ti–Fe$_2$O$_3$ MC photoanode at 1.23 V vs. RHE was 2.5 mA cm$^{-2}$, which is twice and seven times the Fe$_2$O$_3$ MC (1.1 mA cm$^{-2}$) and Fe$_2$O$_3$ SC (0.35 mA cm$^{-2}$) photoanodes, respectively. To the best of our

knowledge, this is the highest value reported thus far for hematite photoanodes without heterojunction or additional surface treatment under back illumination (Supplementary Table 1). By depositing a Co–Pi co-catalyst on the surface, the current density can be further improved to 3.5 mA cm$^{-2}$ (Fig. 3a; Supplementary Fig. 23). The Ti–Fe$_2$O$_3$ MC and Fe$_2$O$_3$ MC photoanodes realize a substantial increase in photocurrent and significant cathodic shift of the onset potential for water oxidation. The Ti–Fe$_2$O$_3$ MC and Fe$_2$O$_3$ MC photoanodes exhibit similar onset potentials of 0.76 V vs. RHE, which is more negative by 0.24 V than that of the Fe$_2$O$_3$ SC photoanode (1.0 V vs. RHE). The observed shift of the onset potential and largely enhanced solar light-responsive current density in MC-based photoanodes imply their superior charge separation and transport abilities[37]. After the deposition of the Co–Pi co-catalyst on the surface, the onset potential further decreases to 0.6 V vs. RHE, indicating that surface recombination is effectively suppressed[38]. To quantitatively investigate the PEC performance of the photoanodes, incident photon-to-current conversion efficiency (IPCE) (Supplementary Note) measurements were performed at 1.23 V vs. RHE in 1.0 M NaOH under back illumination (Fig. 3b). All the samples show photocurrent response to incident light in the wavelength region of 400–620 nm, which matches the bandgap of hematite (~2.1 eV). The Ti–Fe$_2$O$_3$ MC photoanode exhibits substantially enhanced quantum efficiency as compared with other photoanodes in the whole wavelength range. The IPCE of the Ti–Fe$_2$O$_3$ MC photoanode reaches ~31% at the excitation wavelength of 400 nm, which is twice and six times that of Fe$_2$O$_3$ MC (14%) and Fe$_2$O$_3$ SC (5%), respectively. As shown in Fig. 3c, gases were evolved from Ti–Fe$_2$O$_3$ MC photoanode and Pt counter electrode

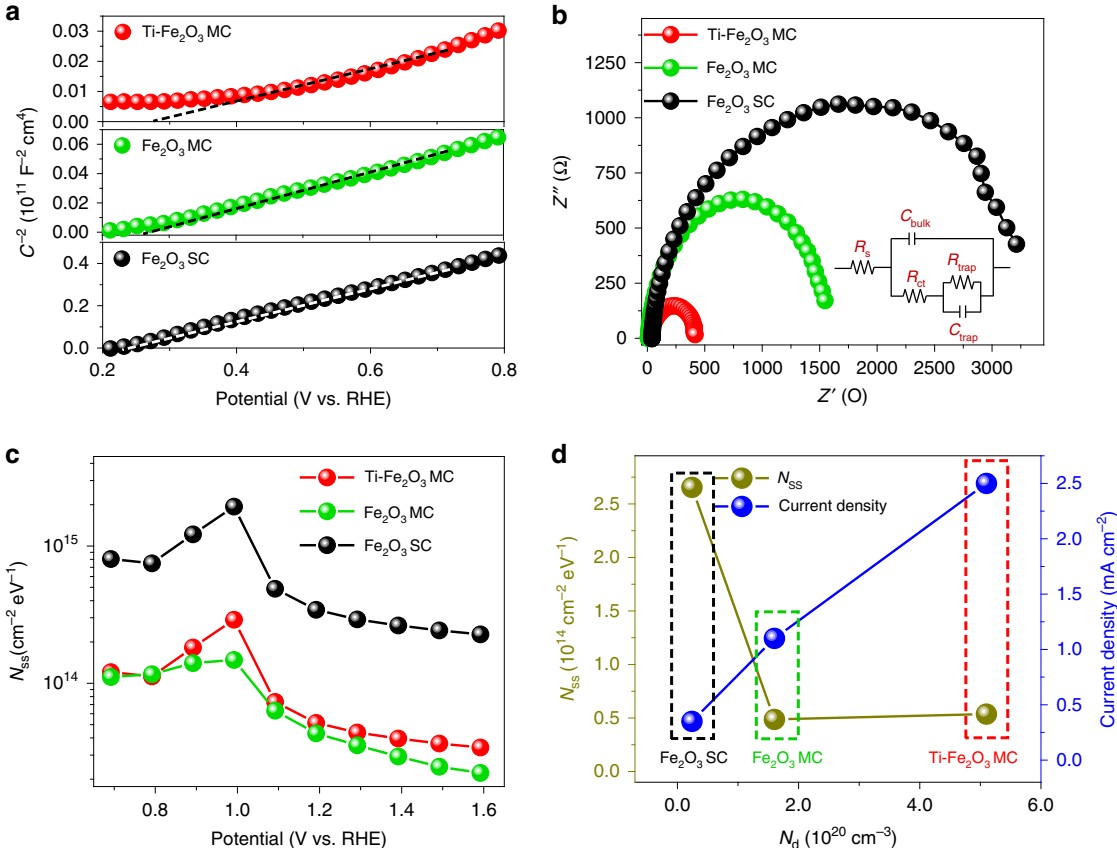

**Fig. 4** Charge transfer efficiencies. **a** Mott–Schottky plots of the samples. Capacitances ($C$s) were determined from electrochemical impedance measurements at a frequency of 10 kHz in the dark. **b** Electrochemical impedance spectra measured for the Ti-$Fe_2O$ MC, $Fe_2O_3$ MC, and $Fe_2O_3$ SC photoanodes. The inset shows the equivalent circuit. **c** The energetic distribution of $N_{SS}$ as a function of the applied potentials. **d** The correlation between $N_{SS}$, $N_d$, and current density at 1.23 V vs. RHE

at 1.23 V vs. RHE with an irradiation area of 0.20 cm$^2$. Both $H_2$ and $O_2$ were linearly produced over 3 h with the stoichiometric ratio (2:1). The Faradaic efficiencies (FEs) for both $H_2$ and $O_2$ are ~97%. The Ti–$Fe_2O_3$ MC photoanode exhibits no obvious current decrease over 24 h, indicating its excellent stability under the operation conditions (Fig. 3d). Figure 3e is the photograph of gas bubbles generated from the Ti–$Fe_2O_3$ MC photoanode and Pt counter electrode at 1.23 V vs. RHE under back illumination.

**Charge transfer efficiencies**. To gain insight into the enhancement of photocurrent in the MC photoanodes, electrochemical impedance measurements were performed at each potential with 10 kHz frequency in the dark. From the Mott–Schottky plots, which represent the changes in capacitance ($C$) against the applied potential, it can be seen that all the samples possess a positive slope (Fig. 4a), which is a characteristic of n-type semiconductors with electrons as majority carriers. The carrier density ($N_d$) can be calculated from the slopes of Mott–Schottky plots (Supplementary Note). The $N_d$ value (Supplementary Table 2) for Ti–$Fe_2O_3$ MC photoanode (5.1 × 10$^{20}$ cm$^{-3}$) is 3.2 and 21 times higher than that of $Fe_2O_3$ MC photoanode (1.6 × 10$^{20}$ cm$^{-3}$) and $Fe_2O_3$ SC photoanode (2.4 × 10$^{19}$ cm$^{-3}$), respectively, suggesting that Ti species play a key role in improving the conductivity and thus efficiently suppress the charge recombination both in the bulk and depletion regions of hematite photoanode. Notably, the electron concentration of the $Fe_2O_3$ MC photoanode is ~6.7 times higher than that of the $Fe_2O_3$ SC photoanode. The higher $N_d$ in $Fe_2O_3$ MCs compared with that of $Fe_2O_3$ SCs originates from

additional electrons, which compensate for the charges induced by the formation of $V_O$ in the crystal, and thus endorse its conductivity[36,39,40]. Therefore, the abundant $V_O$ at the sintered interfaces (i.e., in the bulk) can actually increase the bulk carrier density and thus effectively improve the electrical conductivity.

The charge transport properties of the photoanodes were then investigated by electrochemical impedance spectroscopy (EIS) under back illumination at 1.23 V vs. RHE in 1.0 M NaOH (Fig. 4b). The fitting results according to the equivalent circuit model are summarized in Supplementary Table 2. The series resistance ($R_s$) at the interface between the FTO substrate and hematite layers represents a substantial reduction from ca. 100 Ω cm$^{-2}$ for the unmodified sample to ca. 40 Ω cm$^{-2}$ for the Ti–$Fe_2O_3$ MC sample. This result suggests the possibility that the formation of $TiO_2$ layers on the surface of hematite MCs effectively improves the electron transfer from hematite to the FTO substrate[41]. The charge transfer resistance ($R_{ct}$) increased in the order of Ti–$Fe_2O_3$ MC (367 Ω cm$^{-2}$) < $Fe_2O_3$ MC (576 Ω cm$^{-2}$) < $Fe_2O_3$ SC (2747 Ω cm$^{-2}$), indicating the superior charge mobility in the Ti–$Fe_2O$ MC photoanode due to the increased carrier density by $V_O$ and Ti modification, as well as the highly ordered MC alignment. $R_{trap}$ reflects the charge transfer property at the semiconductor/electrolyte interfaces (SEI). The MC samples exhibited lower $R_{trap}$ values than that of the SC sample, indicating a higher charge transfer efficiency at SEI.

The surface states (SS) are known to strongly influence the charge transfer processes at the SEI[42–50]. To systematically examine the involvement of SS, photoelectrochemical impedance

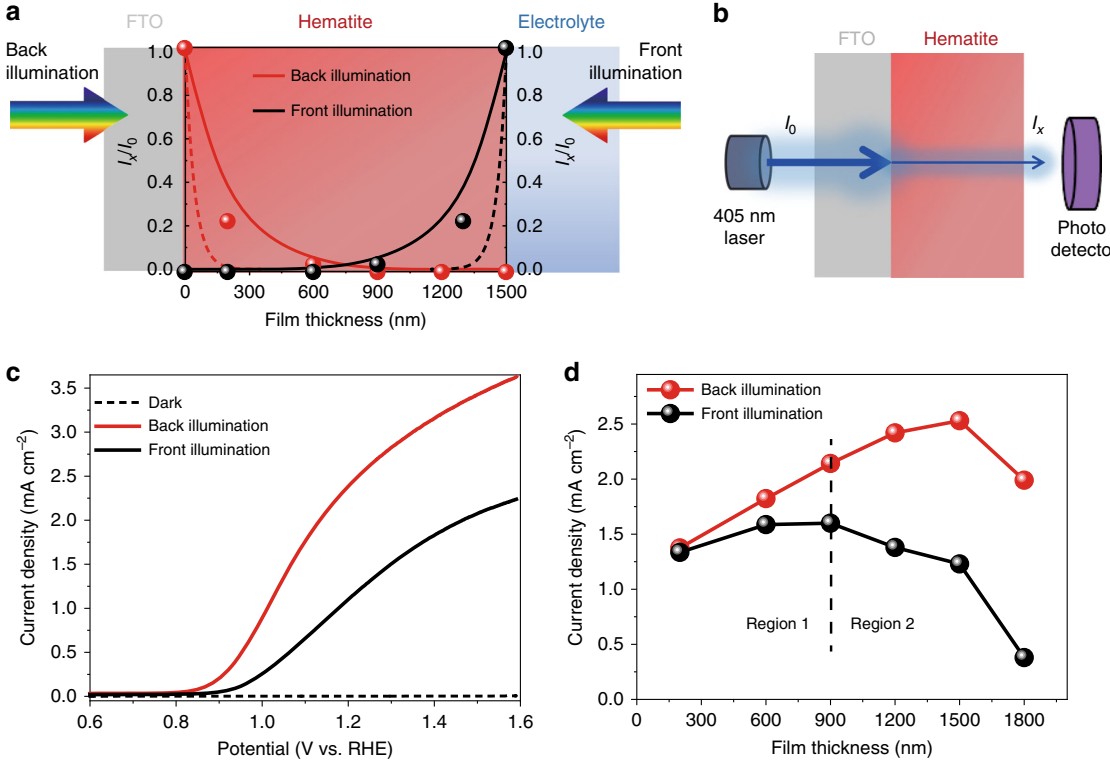

**Fig. 5 Superior back illumination current generation. a** Lambert–Beer law-based simulation of light intensity in hematite film under different illumination modes (dash lines) and the measured light intensity in hematite with different thicknesses (dots) and the corresponding fitted line (solid lines). **b** The method used to measure the light intensity ($I_x/I_0$). **c** The current density–voltage curves of Ti–Fe$_2$O$_3$ MC photoanode (with thickness of 1500 nm) measured under different illumination modes. **d** The current densities at 1.23 V vs. RHE of the Ti–Fe$_2$O$_3$ MC photoanodes with different film thicknesses measured under different illumination modes

spectroscopy (PEIS) measurements were performed at potentials ranging from 0.7 to 1.6 V vs. RHE (Supplementary Fig. 24). The density of mid-gap surface states ($N_{SS}$), which are considered to be related to OH$^-$ and O-terminated surfaces[51], was calculated from $C_{trap}$ (Supplementary Note) and shown in Fig. 4c. The peaks of the energetic distribution of $N_{SS}$ are observed near the formal potential for water oxidation (1.23 V vs. RHE), which decides an equilibration of trap-state energy and hole-accepting species at SEI[44]. The $N_{SS}$ values of MC photoanodes are both lower than that of the SC electrode; this suggests that SS are reduced via oriented attachment of primary nanocrystals, possibly leading to the reduced overpotential and enhanced photovoltage[45,52,53]. These results again support our idea that the hierarchical construction of highly ordered MCs with same crystal orientation effectively improves the PEC performance. In comparison with the Fe$_2$O$_3$ MC photoanode, the slightly increased $N_{SS}$ of Ti-Fe$_2$O$_3$ MC photoanode might be due to the formation of TiO$_2$ overlayer on the surface. The correlation between $N_{SS}$, $N_d$, and current density at 1.23 V vs. RHE is demonstrated in Fig. 4d. The higher $N_d$, which originates from the abundant interfacial V$_O$ and/or Ti incorporation, and lower $N_{SS}$ of MCs would effectively generate higher photocurrents.

**Superior PEC performance under back illumination**. The photocurrent generation of Ti–Fe$_2$O$_3$ MC photoanodes was significantly influenced by the film thickness, as well as illumination mode. Figure 5a illustrates the penetration of excitation light into the hematite film with thickness of 1500 nm under different illumination modes. It is obvious that most of the solar light is absorbed by hematite near the FTO substrate (<300 nm)

according to the Lambert–Beer law (Fig. 5a, dash lines)[54,55]. Since light transmission or scattering might effectively occur in a mesoporous film, the actual intensity ratio of transmitted light ($I_x$) to incident light ($I_0$) (Fig. 5a, dot plot) was measured by the method displayed in Fig. 5b. The fitted results (solid lines in Fig. 5a) show that a much deeper region (up to ~900 nm from the FTO surface) can actually absorb solar light, indicating that more particles can be excited to produce the charge carriers. According to the fact that highest photocurrent density was obtained for the Ti–Fe$_2$O$_3$ MC photoanode with 1500-nm-film thicknesses under back illumination (Fig. 5c), it is presumed that a considerable number of photogenerated electrons diffuse away from the excited region and reach the FTO. Interestingly, the current densities measured under front illumination are much lower than those under back illumination (Fig. 5c).

To understand the charge separation and transport dynamics, the current density of Ti–Fe$_2$O$_3$ MC photoanodes with different film thicknesses are compared under different illumination modes (Fig. 5d; Supplementary Fig. 25). The Ti–Fe$_2$O$_3$ MC photoanodes with different thicknesses all exhibited superior current generation under back illumination in the potential range (0.8 V to 1.6 V) (Supplementary Fig. 25, solid lines). As summarized in Fig. 5d, the current density increased with increasing film thickness from 200 to 900 nm under both illumination modes, which originate from the increase in the number of photons absorbed by the films with larger thicknesses in the region of <900 nm (Region 1). Beyond this region (Region 2, where the film thickness is >900 nm), the photocurrent would be influenced by charge collection efficiency[54]. The EIS results show that Ti–Fe$_2$O$_3$ MC photoanodes with different film thicknesses have lower charge transfer resistances under back illumination than those under front

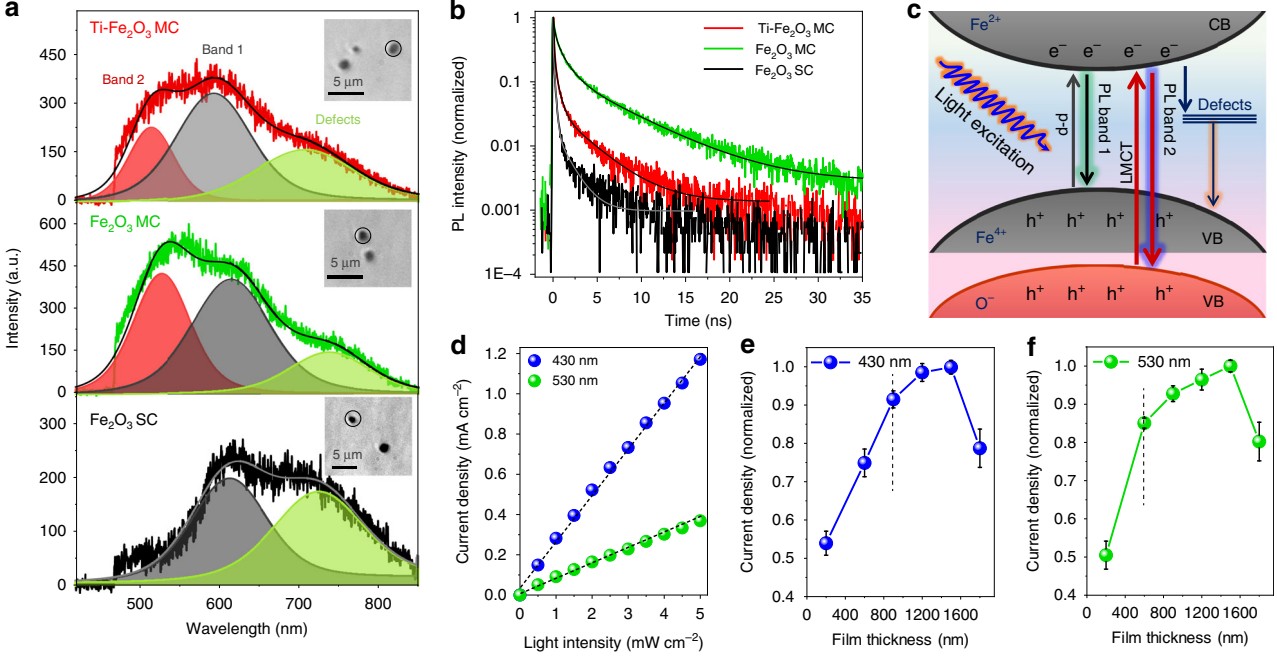

**Fig. 6** Time-resolved PL measurements and wavelength-dependent current generation. **a** Emission spectrum (Inset shows the corresponding optical transmission images) and **b** decay profiles collected from isolated small aggregates of Ti–Fe$_2$O$_3$ MCs, Fe$_2$O$_3$ MCs, and Fe$_2$O$_3$ SCs. **c** The proposed electronic transitions and charge recombination process in MC. **d** Current density at 1.23 V vs. RHE of the Ti–Fe$_2$O$_3$ MC photoanode under monochromatic light illumination (430 nm and 530 nm) from the back side. The normalized current density at 1.23 V vs. RHE of the Ti–Fe$_2$O$_3$ MC photoanodes with different film thicknesses under **e** 430-nm and **f** 530-nm monochromatic light illumination from the back side

illumination (Supplementary Fig. 26). In addition, the charge transfer resistance increases with increasing film thickness, and the lowest charge transfer resistance was observed for the sample with film thickness of 900 nm. On the basis of the above analyses, a plausible mechanism of charge transfer inside the Ti–Fe$_2$O$_3$ MC film under different illumination modes is proposed in Supplementary Fig. 27.

**Charge carrier dynamics**. To elucidate the charge carrier dynamics, time-resolved microspectroscopy measurements were performed. Figure 6a and b depicts the photoluminescence (PL) spectra and decay curves obtained for the particles on quartz cover glasses under 405-nm pulsed laser illumination in ambient air. All the samples exhibit a broad emission band (580–650 nm) centered at ~630 nm (PL band 1), which is close to their bandgaps (~1.9 eV). The broad emission peak centered at a longer wavelength (~720 nm) can be assigned to defect-mediated charge recombination on the surface (Supplementary Table 3). Intriguingly, Ti–Fe$_2$O$_3$ and Fe$_2$O$_3$ MCs display another emission band (510–580 nm) centered at ~560 nm (PL band 2). From the analysis of the PL decay curves, the intensity-weighted average lifetime ($<\tau_{PL}>$) for Fe$_2$O$_3$ SCs was determined to be 0.4 ns (Supplementary Table 4), which is consistent with the reported short lifetime of the photogenerated charges (picosecond time scale). Meanwhile, Ti–Fe$_2$O$_3$ MCs (1.9 ns) and Fe$_2$O$_3$ MCs (4.1 ns) have much longer lifetimes, which are indicative of suppressed charge recombination. The PL decay curves measured with a longpass filter (>593 nm, Supplementary Fig. 28a) show that the PL of Ti–Fe$_2$O$_3$ and Fe$_2$O$_3$ MCs is both quickly quenched (Supplementary Table 4). Therefore, the observed longer lifetimes are likely attributed to PL band 2. This was further proved by the fact that the PL lifetimes measured for MC with a bandpass filter (440–570 nm, Supplementary Fig. 28b) exhibit considerably longer lifetimes (Supplementary Table 4). Furthermore, the

shortened lifetime for Ti–Fe$_2$O$_3$ MCs in comparison with pure Fe$_2$O$_3$ MCs suggests that the charge separation becomes more facile by Ti modification owing to the increased conductivity, thus leading to the best performance.

Based on the above results, the electronic transitions as well as charge-recombination processes in hematite MCs are proposed in Fig. 6c. The MC samples exhibit a much higher content of PL band 2 than that of SCs (Fig. 6a), which corresponds to the higher $V_O$ concentration in MCs (Fig. 2f; Supplementary Fig. 19). The removal of oxygen atoms reduces the coordination number of the neighboring Fe ions, thus leading to strong modification of the local structure. Based on these findings and according to the literature[56], the PL band 2 observed for MC samples might be attributable to radiative recombination of electron–hole pairs generated near $V_O$ via the ligand-to-metal charge transfer (LMCT) transition (O 2p → Fe 3d). It can also be noted that the $V_O$ species not only produce the liberated carriers but also prolong their lifetime. To fully understand the origin and mechanism of PL, however, more work is needed.

Such longer-lived holes would escape from the recombination with the photogenerated electrons, as seen in Ti–Fe$_2$O$_3$ MCs, and thus contribute to the PEC performance. Figure 6d shows the current density of the Ti–Fe$_2$O$_3$ MC photoanode with film thickness of 1500 nm at 1.23 V vs. RHE under back monochromatic light illumination (430 nm and 530 nm) with different light intensities ranging from 0 to 5.0 mW cm$^{-2}$. The corresponding current density–potential curves are given in Supplementary Fig. 29. The current density was linearly enhanced with the increase of light intensity as reported[57]. As the incident light can be fully absorbed by the thick Ti–Fe$_2$O$_3$ MC film, the concentrations of photogenerated electron–hole pairs are almost the same when the sample is excited with same photon fluxes. Despite the fact that the 430-nm light is absorbed closer to the FTO side, i.e., further away from the hematite film/electrolyte interface, as compared with the 530-nm light[55], the current

density obtained by 430-nm excitation is even more than three times higher than that by 530-nm excitation. The higher current density measured by the short-wavelength excitation would be ascribed to the longer hole-diffusion length and higher hole-collection efficiency. To verify this hypothesis, dependences of current generation on the film thickness and light wavelength were further investigated. Supplementary Fig. 30 illustrates the thickness-dependent current generation under monochromatic light (430 nm and 530 nm) illumination with different light intensities. The normalized current densities of different film thicknesses are shown in Fig. 6e. It can be seen that the photocurrent density steadily increases with increasing film thickness from 200 to 900 nm when the sample was excited by 430-nm light in the same manner as simulated solar light illumination (Fig. 5d). In the case of 530-nm excitation, which corresponds to indirect $Fe^{3+}$ d–d transition[56], the result shows a clear increase in current only with the increase of film thickness from 200 to 600 nm (Fig. 6f). Considering the above experimental results, we suggest that the abundant $V_O$ in MC induces the generation of long-lived $O^-$ holes via short-wavelength light excitation, which can largely improve the charge separation and transportation efficiencies between the inner nanocrystal subunits inside the MC.

## Discussion

The separation and collection of the photogenerated charges are driven by the band bending in the depletion layer at the SEI[7]. The width of depletion layer ($W$) can be calculated according the following equation: $W = ((2\varepsilon\varepsilon_0 V_{bi})/(qN_d))^{1/2}$, where $\varepsilon$ is the dielectric constant of hematite (80)[7], $\varepsilon_0$ is the permittivity of free space ($8.854 \times 10^{-12}$ F m$^{-1}$), $V_{bi}$ is the built-in potential which can be calculated by subtracting the flat-band potential from the applied potential, and $q$ is the elementary charge. The $W$ values are calculated to be 4.3, 7.6, and 19.7 nm for Ti–$Fe_2O_3$ MC, $Fe_2O_3$ MC, and $Fe_2O_3$ SC photoanodes, respectively. The $W$ for Ti–$Fe_2O_3$ MC is much smaller than those of $Fe_2O_3$ MC and $Fe_2O_3$ SC, indicating a steeper band bending and a shorter hole drift distance for surface charge collection (Fig. 7a; Supplementary Fig. 31). Because of the excellent penetration of electrolyte into the pores of mesoporous films, the proportion of depletion regions to the entire film should be much higher than that for the SC film (Supplementary Fig. 31). Both the inner pore surface and the outer surface of MCs can contact electrolyte to serve as SE junction regions for charge separation and collection. The

hematite films with thickness >1000 nm and aligned with less conductive [104] direction against the FTO substrate are rarely reported to have a good performance for PEC water oxidation when being irradiated from the back side. The present thick Ti–$Fe_2O_3$ MC photoanodes, however, showed high quantum efficiencies. According to the results of thickness-dependent current generation experiments (Fig. 5d), we can estimate that ~86% of photocurrent was generated in the effective light absorption region (Region 1). Meanwhile, a considerable number of delocalized holes (~14%) generated far away from the film surface (>600 nm) efficiently migrate to Region 2 and drive the water oxidation. The roles of the interfacial $V_O$ and thin $TiO_2$ overlayer in the PEC performance are specifically discussed in Supplementary Discussion with Supplementary Fig. 32.

Based on the above analysis and discussion, we again suggest that assembled MCs with rich interfacial $V_O$ and mesopores have unique properties such as efficient light absorption, superior charge mobility owing to the high carrier density, larger proportion of the depletion region to the entire film, and long-lived holes. These properties are beneficial for charge separation, transfer, and collection and finally enable considerable charges to traverse long distances. According to the PL characteristics and wavelength-dependent photocurrent generation, we can expect that longer-lived $O^-$ holes with excellent mobility are generated with short-wavelength light excitation (Supplementary Discussion), thus enabling highly efficient charge separation and transfer along the intimately connected MCs over 1000 nm distance (Fig. 7b).

Despite the high-performance water splitting under back illumination that has been achieved by the present Ti–$Fe_2O_3$ MC photoanode, the bulk recombination still prevails in the film according to the EIS results. Possible routes for PEC performance improvement include the heterojunction and surface treatment. For instance, developing them into new nanocomposite MCs constructed by two different metal oxides[23] (e.g., $Fe_2O_3$–$MO_x$, M = Ti, Sn, Cr, Ce, etc.) to improve the charge separation and transfer. Considering that the present Ti–$Fe_2O_3$ MCs are aligned along the (104) orientation with relatively low conductivity, we can expect a further improvement of the performance by directly controlling the orientation of MCs in the film. However, the control of crystal orientation on substrate is still a great challenge, which is under investigation by our group.

In conclusion, novel thick hematite MC films (~1500 nm) have been developed for the highly efficient PEC water oxidation under back illumination. Due to the formation of abundant interfacial

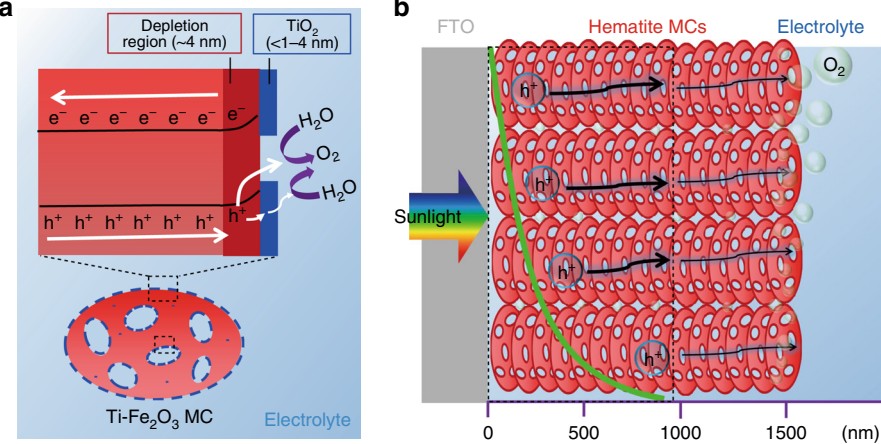

**Fig. 7** Proposed charge carrier dynamics in thick hematite MC film. **a** Illustration of the charge transfer at the SEI of Ti–$Fe_2O_3$ MC. **b** Illustration of the hole transport in the optimized Ti–$Fe_2O_3$ MC photoanode with film thickness of 1500 nm

$V_O$ during partial fusion of interface between the highly ordered nanocrystals within the MC, the charge transfer properties were largely enhanced without significant recombination when compared with the SC film. In addition, the unique mesoporous structures with an extremely high proportion of depletion region and short depletion width would greatly improve the charge separation and collection efficiencies. Moreover, the excitation with shorter wavelength light generates long-lived charges, driving a considerable number of delocalized holes (~14%) to the distance far away from the excited regions along the intimately connected MCs. The present results provide new perspectives to the fields ranging from photocatalysis, electrocatalysis, optoelectronics, and sensing, to energy storage and conversion in terms of interfacial $V_O$ management through the construction of mesocrystal superstructures.

## Methods

**Synthesis of Ti–$Fe_2O_3$ MCs, $Fe_2O_3$ MCs, and $Fe_2O_3$ SCs**. The synthesis of Ti–$Fe_2O_3$ MC, $Fe_2O_3$ MC, and $Fe_2O_3$ SC was based on a simple additive-free solvothermal method. Typically, 1.0 mmol of $Fe(NO_3)_3 \cdot 9H_2O$ (Wako, 99.9%) and 0.1 mmol of $TiF_4$ (Aldrich) were dissolved in a mixture solution containing 40 mL of $N,N$-dimethylformamide (Wako, 99.9%) and 10 mL of methanol by stirring for 20 min. Then the above solution was placed in a 100 mL Teflon-liner autoclave and hydrothermal at 180 °C for 24 h. After naturally cooling, the resulting brown powder was collected by centrifugation at 10,000 rpm, and washed by acetone, water, and methanol for twice, separately. The synthesis of $Fe_2O_3$ MCs was similar to the preparation of Ti–$Fe_2O_3$ MCs by only adding $Fe(NO_3)_3 \cdot 9H_2O$ as a metal precursor in the same mixture solution. The pure $Fe_2O_3$ MC exhibit a bright red color. The $Fe_2O_3$ SC were also prepared as the reference sample only by changing the DMF/methanol volume from 40:10 to 48:2.

**Fabrication of photoanodes by spin coating**. The obtained fresh Ti–$Fe_2O_3$ MCs, $Fe_2O_3$ MCs, and $Fe_2O_3$ SCs were deposited on FTO-coated glasses by spin coating followed by a high-temperature annealing process. Typically, the photocatalyst powders were dispersed in methanol (10 mg mL$^{-1}$) by sonication for 30 min. A piece of dry and cleaned FTO glass (washed by sonication in acetone, methanol, and water for 15 min, separately, and finally immersed in Milli-Q water for use) with a size of $2.2 \times 1.7$ cm$^2$ was placed on a spin-coater. Then, 20 μL of the above suspension was deposited on the FTO glass under a spin-coating speed of 3000 rpm s$^{-1}$ for 10 s. This spin-coating process was repeated for 50 cycles to obtain the optimized film. The thickness of the hematite films was controlled by varying the spin-coating cycles. Finally, the obtained films were annealed at 700 °C for 20 min with a heating speed of 20 °C min$^{-1}$.

**Depositing of Co–Pi co-catalyst on Ti–$Fe_2O_3$ MCs**. Co–Pi co-catalysts were deposited on Ti–$Fe_2O_3$ MC via a photo-assisted electrodeposition method[46]. Typically, a three-electrode cell was utilized with Ti–$Fe_2O_3$ MC film as a working electrode, Pt wire as the counter electrode, and Ag/AgCl as the reference electrode. With simulated sunlight (1 sun, AM 1.5) illumination the Ti-$Fe_2O_3$ MC film, an external bias of 0.4 V vs. Ag/AgCl was applied in a phosphate buffer solution (1.0 M, pH 7) containing 0.5 mM cobalt nitrate. After drying on air, the electrode was carefully washed by water. The amount of Co–Pi co-catalyst was controlled by varying the deposition time from 400 to 700 s.

**Material characterizations**. Powder XRD patterns were recorded on Rigaku Ultima IV with Cu K$_\alpha$ source. SEM images were taken on JEOL JSM-5500 operated at 20 kV. TEM images and EDX mapping images were obtained on a JEM-2100F (JEOL) microscope operated at 200 kV. High-resolution HAADF-STEM images combined with EELS analysis were obtained by a JEM-ARM200F Cold FEG (JEOL) microscope operated at 200 kV, equipped with a Gatan Image Filter Quantum ER. Steady-state UV–visible diffuse reflectance spectrum were collected by a UV–visible spectrophotometer (V-700, JASCO). XPS measurements were performed on a PHI X-tool (ULVAC-PHI). The XPS spectra showed here were calibrated by C1s at 284.8 eV. EPR measurements of the powder hematite samples were carried out by a Bruker EMX spectrometer with an Oxford ESR900 cryostat temperature control system. The microwave power was 6.325 mW, microwave frequency was 9.496 GHz, and temperature was 6 K. $N_2$ adsorption/desorption isotherms at liquid nitrogen temperature (77 K) after degas under vacuum at 150 °C for 6 h were measured using a BELSORP-mini (MicrotracBEL). Synchrotron-based total X-ray total scattering measurements were performed with the incident X-ray energy of $E = 61.4$ keV at BL04B2 beamline in SPring-8, Japan, for studying the local structure from PDF analysis. The hybrid detectors of Ge and CdTe were employed for data collection. The reduced PDF $G(r)$ was obtained by the conventional Fourier transform of the collected data[58].

**PEC measurements**. The PEC data were obtained and analyzed on an electrochemical workstation (ALS, model 608E), and the light source was a compact xenon light source (Asahi Spectra, LAX-C100; 100 mW cm$^{-2}$) equipped with an AM 1.5 filter and calibrated by a silicon photodiode detector (Asahi Spectra, CS-30). In a typical test cell, the fabricated hematite photoanode was applied as the working electrode, a Pt wire was served as the counter electrode, and an Ag/AgCl electrode was used as the reference electrode. The electrolyte solution was 1.0 M NaOH with a pH value of ~13.6. The working area of the hematite electrodes was fixed at 0.49 cm$^2$. The scan rate of the potential was 50 mV s$^{-1}$. All the applied voltage has been converted into the potential vs. RHE via the Nernst equation (Supplementary Note). The gas evolved from the PEC system was analyzed by a gas chromatograph (Shimadzu, GC-8A) equipped with an MS-5A column and a thermal conductivity detector. A xenon light source (Asahi Spectra, MAX-303) coupled with a monochromator (JASCO, CT-10) was used to measure the incident photon to current conversion efficiency (IPCE) data. The Mott–Schottky plots were collected in the dark at a frequency of 10 kHz. The photoelectrochemical impedance spectroscopy (PEIS) under light irradiation were measured in the same electrolyte over a frequency range from 0.05 to 10$^5$ Hz at different applied potentials.

**Time-resolved PL spectroscopy measurements**. The PL properties of individual aggregated particles were recorded using a home-built confocal microscope system based on a Nikon Ti-E inverted fluorescence microscope in ambient air at room temperature. The dry samples deposited on the cleaned quartz cover glass were excited through an objective lens (Nikon, CFI Plan Apo λ 100 × H; numerical aperture 1.45) by a 405 nm pulsed-diode laser (Advanced Laser Diode System, PiL040X) with a pulse width of ~45 ps. Emitted photons were passed through a 100-μm pinhole and directed onto a single-photon avalanche diode (Micro Photon Devices, SPD-050). The obtained signals from the detector were then analyzed by a time-correlated single-photon counting module (Becker & Hickl, SP-130EM).

## Data availability

All the relevant data are available from the authors upon request.

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

## Acknowledgements
Prof. Yasuhiro Kobori (Kobe University) is acknowledged for EPR measurements. KANEKA CORPORATION is acknowledged for financial support. The X-ray total scattering measurements were carried out at SPring-8 under the proposal 2019A2058. This work was partially supported by Nagoya University microstructural characterization platform as a program of "Nanotechnology Platform" of the Ministry of Education, Culture, Sports, Science and Technology (MEXT), Japan, JST PRESTO Grant Number JPMJPR1316, JST A-STEP Grant Number AS3011905T, JSPS KAKENHI Grant Numbers JP18H01944, JP18H04517, and others. Zhujun Zhang thanks to the support from Marubun Research Promotion Foundation.

## Author contributions
Z.Z. and T.T. designed the study, carried out most of the experiments and data analyses, and wrote the paper with input from all authors. I.K. carried out the TEM measurements. H.N. carried out the EPR measurements. S.M. carried out STEM-EELS and multivariate analyses. K.O. and K.S. carried out the synchrotron-based X-ray total scattering and pair distribution function (PDF) analyses.

## Competing interests
The authors declare no competing interests.
