## [Peer Review File · Nature Communications]

Reviewers' comments:

Reviewer #1 (Remarks to the Author):

This article is about photoelectrochemical performance of mesocrystalline MC hematite. The authors study thick films of about 1500 nm and find that photocurrent is higher under back illumination. The MC are composed with nanocrystal subunits of about 30 nm are compared with SC hematite which are composed with nanocrystal of 10 nm. They find that the photocurrent is higher for the MC sample and it can be also improved by Ti doping. They characterized the samples by numerous experimental methods (TEM, XPS, PL...) and try to understand why the MC samples has the better photocurrent. The authors claim that the improvement is due to oxygen vacancies which are introduced in the MC sample during heating at 700 ° in air.

This conclusion is not convincing for several reasons:

1- In previous literature, the minimum temperature to induce reduction of hematite is 1200 degree for heating in air pressure...

2- The authors try to detect oxygen vacancies with O1s XPS. XPS is sensible to only the first nm near the surface and therefore cannot detect oxygen vacancies between crystallite inside a 1500 nm film! Moreover the pic observed at 532 eV is usually attributed to OH-.

3- Moreover for PL measurements the authors attributed the pic around 510-580 nm to oxygen vacancies. This peak is at lower wavelength than the main peak in SC hematite. But in literature the PL peak of oxygen vacancies is usually observed at higher wavelength.

As a conclusion I think that the differences observed between MC and SC sample cannot be simply explained by oxygen vacancies. The arguments of the authors are not convincing. Therefore I do not recommend publication in nature com journal.

Reviewer #2 (Remarks to the Author):

Manuscript ID: NCOMMS-19-06719

Title: "Interfacial oxygen vacancies yielding long-lived holes in mesocrystalline hematite photoanodes"

The paper presents the construction of ordered mesocrystal (MCs) thicker hematite films with good light-to-energy conversion performance under back illumination. The concept of the paper is interesting and the strategy used to prevent charge recombination by increasing the hole-live may add good contribution to the field. Overall the text structure is okay, but there are some grammar mistakes and few sentences that need to be addressed. There are several comments that need to be clarified and will improve the overall manuscript.

1) Since the authors claim the unprecedented back illumination performance some examples reporting what is the state of art for back-illuminated hematite films would be interesting.

2) In the introduction section the authors says: ..."Metal oxide mesocrystals (MCs), which are superstructures of highly ordered nanoparticles assembled with the same crystal orientation, have potentially tunable optical, electronic, and magnetic...", which make me suggest that the use of Mesostructures would be more appropriated than Mesocrystals (MCs).

3) In the introduction section the following sentence is not clear: ..." Therefore, syntheses of hematite MCs and their highly ordered assembly would greatly enhance the charge transfer between nanoparticles in each MC and further between the tightly stacked and intimately attached MCs on the electrode"... The authors does not support the sentence with reports in the literature, which raise a couple of questions: i) Indeed, the crystallographic orientation (in the basal plan) may lead in hematite better electronic conductivity, and in the particular case good crystal interfaces and attachments in between the crystals. However, it creates a large number of interfaces, which is well

established that crystal-crystal interfaces or grain boundaries can act as recombination site. Based on this argument several strategies have been reported in literature to avoid morphology/structure with huge number of interfaces or trying to control/manipulate the crystal interfaces. How the syntheses of mesocrystal solve simultaneously all these problems? It was not clear in a single sentence. In my point of view this a key concept for the entire manuscript.

4) The whole paragraph highlighted below must be better developed. There are unclear sentences from my perspective, such as "fusion of Crystals", and "reconstruction of surfaces caused by solvent". This could be better explained. If "fusion of Crystals" in fact, stands by sintering of the ceramic nanoparticles, then this is a thermally activated process where the great majority of polymers do not withstand. "The stability of MCs is usually improved by using polymer additives, such as double-hydrophilic block co-polymers, to temporarily stabilize the nanoparticles and prevent their fusion to a single crystal.²⁷ However, the removal of additives often requires post-treatment with special solvents, which may cause changes in surface properties as a result of surface reconstruction.^{28,29} Therefore, the development of simple and additive free synthetic methods to fabricate hematite MCs, especially metal element-doped hematite MCs with uniform size, shape, and orientation, is of great significance for the large-scale production of highly active photoanodes, as well as the exploration of model structures to perform detailed investigations into the superstructure-dependent properties in solar energy conversion. "

5) In the sentence authors say..."Further analytical results illustrate that abundant interfacial oxygen vacancies formed during partial fusion at the interfaces between the highly ordered nanoparticles within the MCs can largely increase the carrier density for efficient charge transfer". The word fusion is not appropriated here I would suggest attachment, neck formation (if observed) or sintering. If the "fusion" occurs at the interfaces where the oxygen vacancies will be formed? On other hand, the reduction of number of crystal-crystal interfaces will clearly increases the charge transfer, mainly if the crystal is oriented favoring the anisotropic hematite electronic conductivity.

6) During the synthesis TiF_4 was used as Ti precursor. The films were further annealed at 700C for 20 min. Did the author perform any investigation on residual fluorine arisen from this precursor present in the films? EDX peaks for F can be hidden once O and Fe peaks are close to F peaks. Moreover, it seems that there are non-identified small peaks in the same figure.

7) "The TEM images and corresponding SAED patterns indicate that Ti- Fe_2O_3 MCs maintain their size, morphology, and crystal structure even after high temperature annealing at 700 °C (Figures 1d and S4a,b)." The reason grains are in same size and morphology after thermal treatment is a strong indication there are foreign atoms segregated at the interfaces acting, pinning the movement and / or lowering interfacial energies. Another evidence of segregation of foreign atoms in crystal structure is the minimal or absence of variation on lattice parameter. In fact, this will affect the description of the Ti doped Fe_2O_3 in the manuscript. This analysis must be clarified and some term could be re-thought, such as Ti as modifier. In addition, to evaluate the presence of O- vacancies I would suggest to the authors to perform EELS analysis, this experimental tools can provide evidences of O- vacancy.

8) "The XPS survey spectrum of Ti- Fe_2O_3 MCs clearly indicates Fe 2p, O 1s, and Ti 2p peaks (Figure S8a). As demonstrated in Figure S8b, only the Ti- Fe_2O_3 MC sample exhibited the characteristic peaks at binding energies of 458.6 (Ti 2p_{1/2}) and 464.3 eV (Ti 2p_{3/2}), further proving the successful doping of Ti species into hematite MCs" As XPS technique explores, in its majority, the surfaces of the material, the abundant presence of Ti in the spectra can indicate that Ti is actually segregating at the surfaces and interfaces rather than doping into hematite structure. Although, small portion of Ti could be doping the Fe_2O_3 crystal, but most of the experimental analysis presented in the manuscript shows that most of the added Ti segregated at the hematite crystal interfaces. Indeed, the experimental evidences in the manuscripts does not support the doping effect claimed by the authors.

9) In the following sentence authors says: "The lowest Rct was realized for the Ti- Fe_2O_3 MC

photoanode (180 Ω), which is attributed to the increase in the electrical conductivity by Ti-doping.³⁸⁻⁴⁰ The charge transfer resistance across the semiconductor/electrolyte interface (R_{trap}) reflects the charge transfer property at the interface or in the surface depletion layer." Notably there is a great change in conductivity. Although, Ti not necessarily needs to be inside dopant to improve conductivity. Ti as a segregated atom at the interfaces could also improve electrical conductivity by decreasing resistance at the grain boundaries and hematite film / FTO interfaces. (Please, see references: Nat. Mater. 2013,12, 842 and Phys. Chem. Chem. Phys. 2016, 18, 21780).

10) Figure 4 b, c, d: remove the word The at the beginning. There is an item f instead of item e.

11) "Beyond this region (Region II, where the film thickness is >900 nm), the photocurrent would be influenced by charge collection efficiency". Is the photocurrent not influenced or limited by collection efficiency?

12) Authors sentence: ..."For the MC samples with abundant VO defects, the remaining two electrons from the O_2^- site will occupy available Fe 3d states of the neighboring Fe ions. This structural change shortens the local Fe-O bonds^{44,50} and leads to greater Fe d and O p orbital hybridization, which correspond to the chemical shifts verified by the XPS analysis (Figures 3d and S8c). Therefore, the ligand-to-metal charge transfer (LMCT) transition ($\text{O } 2p \rightarrow \text{Fe } 3d$) centered at 3.2 eV⁵⁵ can effectively occur when being excited by short-wavelength light..." This explains the shift for Fe_2O_3 MCs, however, there should be a different bond behavior for Ti doped situation both in emission band and PL decay. If it is not the case, then Ti is not influencing the Fe_2O_3 structure, which supports the idea that Ti is mostly segregated at the interfaces of Fe_2O_3 . This needs to be clarified by the authors. From my perspective the data could all be explained with segregation.

13) EPR measurement confirmed the presence of Fe^{+2} . The presence of high amount Fe^{+2} may act trapping electrons instead of favor the electron hopping mechanism in hematite. How the presence of Fe^{+2} could positively affect the charge transport? In addition, several papers showed that the presence of Fe^{+2} in hematite provide inefficient surface for chemical reaction.

Reviewer #3 (Remarks to the Author):

In this work, the authors reported the preparation of novel thick hematite films (~ 1500 nm) constructed by highly ordered and intimately attached hematite mesocrystals (MCs) for highly efficient PEC water oxidation. The study is somehow comprehensive and well organized. The authors attributed the enhanced PEC water oxidation to the high carrier density derived from oxygen vacancies. But the authors neglect the role of surface states presented in the hematite/electrolyte interfaces, which is very important for the charge transfer during PEC water splitting. Additionally, there are some technical issues need to be addressed. Thus, I recommend it to be major revised before published on "Nature Communications". My detail suggestions are as follows:

(1) The TEM images in Figure 1a and 1d reveal that discus-like structure is composed of some nanoparticles, indicating the whole discus-like structure is not a single crystalline. But the authors show a SAED with a single set of diffraction pattern, which is an indicative of single crystalline? The authors should indicate the accurate region where they got the SAED data!

(2) The fusion interfaces with distorted lattice fringes in Figure 1d, meaning abundant defects, which could be dislocations, twins, vacancy etc. There is no directly TEM data can demonstrate the fusion interfaces is derived from oxygen vacancies. Since the authors relate the observed fusion interfaces with the oxygen vacancies in the discussion part, in order to verify the presence of oxygen vacancies, the authors should use the relative EELS composition map to quantitative the oxygen elements content and use spherical aberration corrected STEM-ABF to directly observe the oxygen atomic column. The author may refer to these references:

[1] Synergistic Effects in 3D Honeycomb-like Hematite Nanoflakes/Branched Polypyrrole Nanoleaves

Heterostructures as High-Performance Negative Electrodes for Asymmetric Supercapacitors, *Nano Energy*, 2016, 22, 189-201.

[2] High-yield Synthesis and Optical Properties of g-C₃N₄, *Nanoscale*, 2015, 7, 12343-12350.

[3] Polarity assignment in ZnTe, GaAs, ZnO and GaN-AlN nanowires from direct dumbbell analysis, *Nano Letters*, 2012, 12, 2579-2586.

(3) The grain sizes of Ti-Fe₂O₃ and Fe₂O₃ MCs are about several hundred nanometres, which is distinctly different from the value (30 nm) obtained from XRD. The authors should explain it!

(4) In Figure S1, except the discus-like structure, there is also some rugger-like structure; is it a residual or a mixture of discus-like structure and rugger-like structure?

(5) In table S1, why the Rs reduces so much in the case of Ti-Fe₂O₃ MCs? Where is the data of Ctrap? Where is the flatland data of these three electrodes and how does they change in these three electrodes? The authors should present all the important data, including flatland and Ctrap from EIS and Mott-Schottky plots in the manuscript and SI!

(6) As show in Fig. 5C, there are always different defects presented at the surface of hematite, mediating the charge transfer at the hematite/electrolyte interfaces. The authors solely attributed the enhanced PEC performance to the enhanced electronic conductivity resulted from the improved oxygen vacancy but neglected the possible change of the surface states in hematite. Is there any change of the surface states in hematite upon Ti doping and nanostucturing? What is the role of the surface states in hematite during the PEC water splitting? The authors should refer to books from Juan Bisquert: Photoelectrochemical solar fuel production-from basic principles to advanced devices, [M]. 2016, Berlin: Springer and the following references:

[1] Water oxidation at hematite photoelectrodes: The role of surface states, *J. Am. Chem. Soc.*, 2012, 134, 4294-4302.

[3] Understanding the role of underlayers and overlayers in thin film hematite photoanodes, *Adv. Funct. Mater.* 2014, 24, 7681-7688.

[4] Electrochemical and photoelectrochemical investigation of water oxidation with hematite electrodes, *Energy Environ. Sci.*, 2012, 5, 7626-7636.

[5] Water oxidation on hematite photoelectrodes: insight into the nature of surface states through in situ spectroelectrochemistry, *J. Phys. Chem. C*, 2014, 118, 10393-10399.

[6] Enhanced water splitting efficiency through selective surface state removal, *J. Phys. Chem. Lett.*, 2014, 5, 1522-1526.

[7] What do you do, Titanium? Insight into the Role of Titanium Oxide as Water Oxidation Promoter in Hematite-based Photoanodes, *Energy & Environmental Science*, 2015, 8, 3242-3254.

[8] Determination of photoelectrochemical water oxidation intermediates on haematite electrode surfaces using operando infrared spectroscopy, *Nat. Chem.*, 2016, 8, 778-783.

[9] Enhanced Photoelectrochemical Water Splitting of Hematite Multilayer Nanowires Photoanode with Tuning Surface State via Bottom-up Interfacial Engineering, *Energy & Environmental Science*, 2017, 10, 2124-2136.

The authors should systematically monitor the donor density and surface states evolution simultaneously via PEIS and EIS under dark. The authors should clearly clarify the contributions of enhanced electronic conductivity and changed surface states to the enhanced PEC water splitting, respectively!

(7) In Figure 4. f, The proposed diffusion processes of charge carriers under different illumination modes should be Figure 4. e, The proposed diffusion processes of charge carriers under different illumination modes.

(8) There seems to be no data reporting the detection or quantification of oxygen. As this is a paper about photoelectrochemical water oxidation (presumably to produce oxygen), that seems to be an omission. I admit that it is highly likely that oxygen is the main product, but as it stands an unkind referee could argue that the lack of this oxygen data undermines the conclusions of the paper. I

therefore suggest that some oxygen detection data are included, and a Faradaic yield for oxygen quoted.

Responses to the comments from Reviewer #1:

This article is about photoelectrochemical performance of mesocrystalline MC hematite. The authors study thick films of about 1500 nm and find that photocurrent is higher under back illumination. The MC are composed with nanocrystal subunits of about 30 nm are compared with SC hematite which are composed with nanocrystal of 10 nm. They find that the photocurrent is higher for the MC sample and it can be also improved by Ti doping. They characterized the samples by numerous experimental methods (TEM, XPS, PL...) and try to understand why the MC samples has the better photocurrent. The authors claim that the improvement is due to oxygen vacancies which are introduced in the MC sample during heating at 700 ° in air. This conclusion is not convincing for several reasons:

The authors express sincere thanks to the reviewer for his/her valuable comments and constructive suggestions.

1- In previous literature, the minimum temperature to induce reduction of hematite is 1200 degree for heating in air pressure...

Reply: We strongly agree with the reviewer that the reduction of hematite in air requires high temperature. Actually, when the Fe₂O₃ SC reference sample was heated at the same temperature (700 °C) in air, oxygen vacancy (V_O) did not form as shown in Figs. 2g and h. Therefore, this result can strongly support our hypothesis that abundant V_O formed at the sintered interfaces inside MCs. To obtain more evidence for effective generation of V_O in the MC samples, we carried out STEM-EELS measurements (Figs. 2d-f). Based on the TEM, EELS, EPR, and XPS results, we propose that the partial sintering of highly ordered interfaces between the nanocrystal subunits inside MCs at high temperature could leave small interior spaces with oxygen deficiency (i.e., oxygen deficient interfaces), and thus oxygen atoms in the hematite lattice (Fe-O) more easily create V_O ($\text{Fe-O} + \text{Fe-O} \rightarrow 2\text{Fe-V}_\text{O} + \text{O}_2$) (Fig. 2i) due to the lower formation energy of V_O in metal oxides compared with cation interstitials.

2- The authors try to detect oxygen vacancies with O1s XPS. XPS is sensible to only the first nm near the surface and therefore cannot detect oxygen vacancies between crystallite inside a 1500 nm film! Moreover the pic observed at 532 eV is usually attributed to OH-

Reply: We agree with his/her comment that XPS is sensitive to only the first nm near the surface. The sample for XPS measurements was the powder collected from the film but not the film itself. As the samples possess abundant pores (Figs. 1d, f, and 2e) due to thermal sintering of the

inner nanocrystal subunits, a part of sintered interfaces would be located at the regions near the edges of the pores, which is possible to be detected by XPS.

The careful analysis of the O1s XPS spectra verifies the presence of three peaks. The peak at around 531.9 eV can be assigned to OH⁻ group as the reviewer suggested. The literature reported that the detected peak was due to the hydroxyl groups (OH⁻) bonded to the metal cations (Fe-OH) in the oxygen deficient region. Therefore, a higher concentration of OH⁻ groups might be due to a higher concentration of oxygen vacancies. Accordingly, we have revised the expression as “Because the formation of V_O alters the coordination of O atoms and chemical valance state of Fe cations in hematite,³⁸⁻⁴⁰ the oxygen species at the surface and exposed interfaces were characterized by the O 1s XPS spectra. As shown in Fig. 2g, peaks with binding energies at approximately 530.2, 531.9, and 533.3 eV were observed for the samples, corresponding to the O²⁻ species in the hematite lattice (O_L), hydroxyl groups (OH⁻) bonded to the metal cations (Fe-OH) in the oxygen deficient region (O_V), and chemisorbed or dissociated oxygen species from the H₂O molecules (O_C), respectively.^{38”} in the revised manuscript (page 12).

To prove the formation of abundant interfacial V_O, high resolution HAADF-STEM measurements combined with EELS maps were carried out at a typical region with sintered interfaces (Figs. 2d, e). The valence states of iron were identified by Fe-L_{2,3} EELS data (Fig. 2f) and separately visualized in Fig. 2e. Fe²⁺ (i.e., V_O) are mostly distributed in the dark zone where the distorted lattice fringes are seen (Fig. 2d), while Fe³⁺ are mostly distributed in the bright zone where clear lattice fringes are seen. More results and discussion were added in the revised manuscript (pages 10-11).

3- Moreover for PL measurements the authors attributed the pic around 510-580 nm to oxygen vacancies. This peak is at lower wavelength than the main peak in SC hematite. But in literature the PL peak of oxygen vacancies is usually observed at higher wavelength.

As a conclusion I think that the differences observed between MC and SC sample cannot be simply explained by oxygen vacancies. The arguments of the authors are not convincing. Therefore I do not recommend publication in nature com journal.

Reply: Thank you very much for the comment. The PL emission band at 510-580 nm can only be detected for the MC samples that contain abundant interfacial oxygen vacancies as proved by STEM-EELS, EPR, and XPS results. The peak centered at ~560 nm (PL band II) cannot be directly assigned to the oxygen vacancies as well as impurities. Actually, the broad emission band centered at the longer wavelength (~720 nm) can be assigned to defect-mediated charge recombination on the surface, as mentioned in the original version. Similar results were also

reported in the literature, as summarized in Supplementary Table 3. The new emission band centered at ~560 nm (PL band II) is highly correlated with the oxygen vacancies, though its origin is still not fully understood. To avoid any confusion, we have clarified in the revised manuscript (pages 22-24) as “For the MC samples with abundant V_O defects, the remaining two electrons from the O^{2-} site will occupy available Fe 3d states of the neighboring Fe ions. This structural change may shorten the local Fe–O bonds^{40,43} and lead to greater Fe d and O p orbital hybridization, which correspond to the chemical shifts verified by the XPS analysis (Fig. 2g and Supplementary Fig. 9c). Therefore, the ligand-to-metal charge transfer (LMCT) transition ($O\ 2p \rightarrow Fe\ 3d$) centered at 3.2 eV⁶³ can effectively occur when being excited by short-wavelength light. A recent study employing time-resolved microwave conductivity suggested that the 355 nm laser excitation of pure and metal-doped hematite films produces more mobile charges than excitation with 532 nm light that is not well matched with the LMCT transition.⁶⁴ Due to the low concentration of V_O defects in Fe_2O_3 SCs, the distance of local Fe–O bonds remains basically unchanged and thus this transition is not effective. Accordingly, the PL band II observed for MC samples can be assigned to radiative recombination of (shallowly trapped) electron (Fe^{2+})-hole (O^-) pairs (Fig. 6c). The Fe_2O_3 MCs have a higher content of PL band II than that of Ti- Fe_2O_3 MCs (Fig. 6a), which is consistent with the higher V_O concentration in Fe_2O_3 MCs (Fig. 2g). Furthermore, as shown in Fig. 6b, the shortened lifetime for Ti- Fe_2O_3 MCs in comparison with pure Fe_2O_3 MCs suggests that the charge separation becomes more facile by Ti modification owing to the increased conductivity, thus leading to the best performance. Based on the above results, the electronic transitions as well as charge recombination processes in hematite MC are proposed as Fig. 6c shows. Recently, the excitation-wavelength-dependent lifetime of the photoexcited electrons was explained in terms of polaron-hopping theory.⁶⁵ The higher excitation energy provides more excess energy to the lattice, i.e., fewer polarons being formed by the non-thermal phonon bath, and thus the hopping rate of the polarons in equilibrium with mobile carriers will increase, resulting in the increased hopping radius and lifetime of carriers. However, this model cannot solely explain our finding that the nanosecond PL lifetimes were observed only for the MC samples. Therefore, it can be suggested that the V_O species not only produce the liberated carriers but also the long-lived O^- holes via the enhanced LMCT transition. Such longer-lived holes would escape from the recombination with the photogenerated electrons, as seen in Ti- Fe_2O_3 MCs, and thus contribute to the PEC performance. To fully understand the origin and mechanism of PL, however, more work is needed.”

Responses to the comments from Reviewer #2:

The paper presents the construction of ordered mesocrystal (MCs) thick hematite films with good light-to-energy conversion performance under back illumination. The concept of the paper is interesting and the strategy used to prevent charge recombination by increasing the hole-life may add good contribution to the field. Overall the text structure is okay, but there are some grammar mistakes and few sentences that need to be addressed. There are several comments that need to be clarified and will improve the overall manuscript.

The authors express sincere thanks to the reviewer for his/her valuable comments and constructive suggestions.

1) Since the authors claim the unprecedented back illumination performance some examples reporting what is the state of art for back-illuminated hematite films would be interesting.

Reply: According to the reviewer's comment, we have provided Supplementary Table 1 showing the reported back illumination PEC performance of hematite-based photoanodes without heterojunction and surface treatment.

2) In the introduction section the authors says: ...“Metal oxide mesocrystals (MCs), which are superstructures of highly ordered nanoparticles assembled with the same crystal orientation, have potentially tunable optical, electronic, and magnetic...”, which make me suggest that the use of Mesosstructures would be more appropriated than Mesocrystals (MCs).

Reply: We appreciate for the reviewer's comment and suggestion. To reduce the concept of ambiguity, we revised the sentence “Metal oxide mesocrystals (MCs), which are superstructures of highly ordered nanoparticles assembled with the same crystal orientation, have potentially tunable optical, electronic, and magnetic...” as “Mesocrystals (MCs) are superstructures of nanoparticles with a specific preferable mutual orientation.” in the revised manuscript (page 3).

3) In the introduction section the following sentence is not clear: ...“ Therefore, syntheses of hematite MCs and their highly ordered assembly would greatly enhance the charge transfer between nanoparticles in each MC and further between the tightly stacked and intimately attached MCs on the electrode”... The authors does not support the sentence with reports in the literature, which raise a couple of questions: i) Indeed, the crystallographic orientation (in the basal plan) may lead in hematite better electronic conductivity, and in the particular case good crystal interfaces and attachments in between the crystals. However, it creates a large number of interfaces, which is well established that crystal-crystal interfaces or grain boundaries can act as

recombination site. Based on this argument several strategies have been reported in literature to avoid morphology/structure with huge number of interfaces or trying to control/manipulate the crystal interfaces. How the syntheses of mesocrystal solve simultaneously all these problems? It was not clear in a single sentence. In my point of view this a key concept for the entire manuscript.

Reply: We appreciate for this critical comment. To address this issue, we revised the sentence “Therefore, syntheses of hematite MCs and their highly ordered assembly would greatly enhance the charge transfer between nanoparticles in each MC and further between the tightly stacked and intimately attached MCs on the electrode” as “In addition, the crystal-crystal interfaces or grain boundaries may act as recombination sites due to the lattice mismatch in crystals with different orientation and thus limit charge transport.²⁰ Mesocrystals (MCs) are superstructures of nanoparticles with a specific preferable mutual orientation.^{21,22} Tachikawa et al. demonstrated that MCs of metal oxides, such as TiO₂ and SrTiO₃, have superior efficiencies in charge separation and transport between primary nanocrystals compared with conventional nanocrystal systems owing to their highly ordered structures.²³⁻²⁷ Besides, the interfacial atomic structures between facing nanocrystal subunits inside MC might be partly adjusted to reduce the recombination sites, possibly resulting in the improvement of intergranular electronic conductivity. Therefore, the hieratical assembly of MCs with minimum disorders and appropriate adjustment of the interface may further improve the PEC performance.” in the revised manuscript (pages 3-4).

4) The whole paragraph highlighted below must be better developed. There are unclear sentences from my perspective, such as “fusion of Crystals”, and “reconstruction of surfaces caused by solvent”. This could be better explained. If “fusion of Crystals” in fact, stands by sintering of the ceramic nanoparticles, then this is a thermally activated process where the great majority of polymers do not withstand. “The stability of MCs is usually improved by using polymer additives, such as double-hydrophilic block co-polymers, to temporarily stabilize the nanoparticles and prevent their fusion to a single crystal.²⁷ However, the removal of additives often requires post-treatment with special solvents, which may cause changes in surface properties as a result of surface reconstruction.^{28,29} Therefore, the development of simple and additive free synthetic methods to fabricate hematite MCs, especially metal element-doped hematite MCs with uniform size, shape, and orientation, is of great significance for the large-scale production of highly active photoanodes, as well as the exploration of model structures to perform detailed investigations into the superstructure-dependent properties in solar energy conversion.”

Reply: We greatly appreciate for the reviewer's suggestion. For the sake of streamlining and readability, we delated the paragraph "The stability of MCs is usually improved by using polymer additives, ...".

5) In the sentence authors say...“Further analytical results illustrate that abundant interfacial oxygen vacancies formed during partial fusion at the interfaces between the highly ordered nanoparticles within the MCs can largely increase the carrier density for efficient charge transfer”. The word fusion is not appropriated here I would suggest attachment, neck formation (if observed) or sintering. If the “fusion” occurs at the interfaces where the oxygen vacancies will be formed? On other hand, the reduction of number of crystal-crystal interfaces will clearly increases the charge transfer, mainly if the crystal is oriented favoring the anisotropic hematite electronic conductivity.

Reply: We greatly appreciate for the reviewer's suggestions. We have changed the “fusion” with “sintering” in the whole manuscript. We also agree with the comment that the reduction of number of crystal-crystal interfaces will increase the charge transfer. If a more favorable crystallographic direction (e.g., [110]) of the MCs is oriented perpendicular to the FTO surface, the electronic conductivity and PEC performance might be more improved. Such a study is underway in our group.

6) During the synthesis TiF4 was used as Ti precursor. The films were further annealed at 700C for 20 min. Did the author perform any investigation on residual fluorine arisen from this precursor present in the films? EDX peaks for F can be hidden once O and Fe peaks are close to F peaks. Moreover, it seems that there are non-identified small peaks in the same figure.

Reply: We greatly appreciate for the reviewer's comment. Actually, as synthesized Ti-Fe₂O₃ MCs were thoroughly washed with methanol and water to remove any remaining residuals. To confirm the possibility of fluorine residuals on the surface, we carried out the XPS F1s analysis as the following figure shows. F was not detected in both the F1s and survey spectrum (Supplementary Fig. 9). The non-identified small peaks are Cu originated from the Cu substrate used for TEM measurements. We have identified the peaks in the revised manuscript (Supplementary Fig. 3).

7) “The TEM images and corresponding SAED patterns indicate that Ti-Fe₂O₃ MCs maintain their size, morphology, and crystal structure even after high temperature annealing at 700 °C (Figures 1d and S4a,b).” The reason grains are in same size and morphology after thermal treatment is a strong indication there are foreign atoms segregated at the interfaces acting, pinning the movement and / or lowering interfacial energies. Another evidence of segregation of foreign atoms in crystal structure is the minimal or absence of variation on lattice parameter. In fact, this will affect the description of the Ti doped Fe₂O₃ in the manuscript. This analysis must be clarified and some term could be re-thought, such as Ti as modifier. In addition, to evaluate the presence of O- vacancies I would suggest to the authors to perform EELS analysis, this experimental tools can provide evidences of O- vacancy.

Reply: We greatly appreciate for the important suggestions. We have clearly determined the distribution and concentration of Ti and oxygen in the crystal by STEM-EELS measurements. The manuscript was revised as “HAADF-STEM with electron energy loss spectroscopy (EELS) measurements can give useful information about the local composition, structure, and chemical states of the samples.³²⁻³⁴ Fig. 2a shows the HAADF-STEM image of a typical Ti-Fe₂O₃ MC particle after being annealed at 700 °C in air. The corresponding EELS composition map is given in panel b. It is obvious that most of the Ti species are segregated to the outer surface and edges of the pores. To further analyze the distribution of the elements in the crystal, EEL spectra were acquired in different regions (Fig. 2a). As shown in Fig. 2c, only Ti-L_{2,3} and O-K signals were detected on the edge of the MC particle (Region 1 in Fig. 2a), suggesting the presence of TiO₂ overlayer. The thicknesses of TiO₂ layers are estimated from <1 to approximately 4 nm (Fig. 2b and Supplementary Fig. 11). It is noteworthy that TiO₂ (most probably rutile phase) grew on the surface of hematite as suggested by the high-resolution HAADF-STEM image, the

corresponding fast Fourier transform (FFT) pattern (Supplementary Fig. 11), and the XRD peak attributed to rutile (110) (Supplementary Fig. 12). On the smooth surface (Region 2 in Fig. 2a), Ti-L_{2,3} signals are very weak, implying that a small portion of Ti⁴⁺ ions were doped into the crystal. A higher concentration of Ti⁴⁺ ions were observed near the edges of the pores (Region 3 in Fig. 2a), again indicating the formation of thin TiO₂ layers.

The synchrotron-based X-ray total scattering and pair distribution function (PDF) analyses were further performed to investigate the local structural changes in MCs (Supplementary Fig. 13). For Fe₂O₃ MCs, no apparent difference in Fe–O and Fe–Fe bond distances was observed between the samples before and after the annealing. This is probably due to the fact that the amounts of V_O in annealed MCs are still too small to be detected. Whereas, the Ti modification results in a significant intensity decrease and broadening of the peaks. After the annealing, the spectral features are almost restored to those of pure hematite, indicating that most Ti ions are segmented from the bulk to the surface while leaving a trace as dopant.

To prove the formation of abundant interfacial V_O, high-resolution HAADF-STEM measurements combined with EELS maps were carried out at typical region with sintered interfaces (Figs. 2d, e). The valence states of iron were identified by Fe-L_{2,3} EEL spectra (Fig. 2f) and separately visualized in Fig. 2e using multiple linear least-square fit to the spectrum image data with the reference spectra extracted from the well-defined areas. Fe²⁺ ions (i.e., V_O)³⁵ are mostly distributed in the dark zone where the distorted lattice fringes are seen (Fig. 2d), while Fe³⁺ ions are distributed in the bright zone where clear lattice fringes are seen. The multivariate analysis³⁶ of spectrum imaging data was further performed to clarify the associations between the specific elements (Supplementary Fig. 14). The analytical results imply that Fe²⁺ species originate not only from V_O at the interface in MCs (Fig. 2e), but also from Fe_{2-x}Ti_xO₃ (e.g., ilmenite) on the surface. The latter is considered to be present at the interface between hematite and TiO₂ (Supplementary Fig. 11) and play a key role in the formation of the TiO₂ overlayer during the annealing.³⁷.

8) “The XPS survey spectrum of Ti-Fe₂O₃ MCs clearly indicates Fe 2p, O 1s, and Ti 2p peaks (Figure S8a). As demonstrated in Figure S8b, only the Ti-Fe₂O₃ MC sample exhibited the characteristic peaks at binding energies of 458.6 (Ti 2p_{1/2}) and 464.3 eV (Ti 2p_{3/2}), further proving the successful doping of Ti species into hematite MCs” As XPS technique explores, in its majority, the surfaces of the material, the abundant presence of Ti in the spectra can indicate that Ti is actually segregating at the surfaces and interfaces rather than doping into hematite structure. Although, small portion of Ti could be doping the Fe₂O₃ crystal, but most of the experimental analysis presented in the manuscript shows that most of the added Ti segregated at the hematite crystal interfaces. Indeed, the experimental evidences in the manuscripts does not

support the doping effect claimed by the authors.

Reply: We agree with reviewer's comment. As shown in Supplementary Figs. 10, the XPS depth profiles obtained for Ti-Fe₂O₃ MCs revealed that most of Ti atoms are actually segregated to the surfaces, and small concentration of Ti ions were doped into the hematite crystal. In addition, the synchrotron-based total X-ray scattering and pair distribution function (PDF) analyses were performed to investigate the local structural changes in MCs (Supplementary Figs. 13). For pure hematite MCs, no detectable difference was observed before and after high temperature annealing. This is probably due to the fact that the amount of oxygen vacancies is too small to be detected. Whereas, the Ti modification results in a clear decrease in peak intensity and peak broadening. After high temperature annealing, the intensities are almost restored to those of pure hematite, indicating that most Ti atoms are segmented from the bulk to the surface. Therefore, the term "Ti-doped Fe₂O₃ MCs" was changed to "Ti-modified Fe₂O₃ MCs" in the whole manuscript.

Based on the above additional experiments, we have clarified the element distribution in the Ti-Fe₂O₃ MCs by means of HAADF-STEM with EELS. The results given in Fig. 2a-c clearly show that most of the Ti species are segregated to the outer surface and edges of the pores. The thicknesses of Ti layers are estimated from <1 to 4 nm (Fig. 2b and Supplementary Fig. 11). Rutile-phase TiO₂ is well grown on the surface of hematite as shown by the high resolution HAADF-STEM image and the corresponding fast Fourier transform (FFT) patterns (Supplementary Fig. 11). To further analyze the distribution of the elements in the crystal, EEL spectra were acquired and shown in Fig. 2c. Only Ti-L and O-K signals were detected on the outer surface of the MC particle (Region 1 in Fig. 2a), further proving the presence of the TiO₂ overlayer. In the bulk of the MC (Region 2 in Fig. 2a), Ti-L signals are very weak, implying that a small portion of Ti⁴⁺ ions were doped into the crystal. A higher concentration of Ti⁴⁺ ions were observed near the edges of the pores (Region 3 in Fig. 2a), again indicating the formation of thin TiO₂ layer. We added those results and discussion in the revised manuscript (pages 9-11) and SI.

9) In the following sentence authors says: "The lowest R_{ct} was realized for the Ti-Fe₂O₃ MC photoanode (180 Ω), which is attributed to the increase in the electrical conductivity by Ti-doping.³⁸⁻⁴⁰ The charge transfer resistance across the semiconductor/electrolyte interface (R_{trap}) reflects the charge transfer property at the interface or in the surface depletion layer." Notably there is a great change in conductivity. Although, Ti not necessarily needs to be inside dopant to improve conductivity. Ti as a segregated atom at the interfaces could also improve electrical conductivity by decreasing resistance at the grain boundaries and hematite film / FTO interfaces. (Please, see references: Nat. Mater. 2013,12, 842 and Phys. Chem. Chem. Phys. 2016,

18, 21780).

Reply: We agree with the reviewer's comment. To systemically study the charge transfer in the electrode, photoelectrochemical impedance spectroscopy (PEIS) measurements were performed as shown in Fig. 4 and Supplementary Fig. 19. The corresponding results are added with discussion in the revised manuscript (pages 18-19) and can support the comment from the reviewer.

10) Figure 4 b, c, d: remove the word The at the beginning. There is an item f instead of item e.

Reply: Thanks for the reviewer's comments. We have corrected the legends in the revised manuscript.

11) "Beyond this region (Region II, where the film thickness is >900 nm), the photocurrent would be influenced by charge collection efficiency". Is the photocurrent not influenced or limited by collection efficiency?

Reply: Thanks to the reviewer's comment. As explained in the manuscript, light can be absorbed with a film thickness of ~900 nm and the photocurrent lineally increased with increasing film thickness in this region. Therefore, the photocurrent generation is highly correlated with light absorption efficiency in this region. Beyond the light absorption region (Region II), photoinduced charge separation and transfer usually do not occur in Region II, and therefore the photocurrent generated in Region II is due to charge transfer from Region I to Region II. Thus, the increased photocurrent in Region II is highly dependent on the charge collection efficiency in this region.

12) Authors sentence: ..."For the MC samples with abundant VO defects, the remaining two electrons from the O²⁻ site will occupy available Fe 3d states of the neighboring Fe ions. This structural change shortens the local Fe-O bonds^{44,50} and leads to greater Fe d and O p orbital hybridization, which correspond to the chemical shifts verified by the XPS analysis (Figures 3d and S8c). Therefore, the ligand-to-metal charge transfer (LMCT) transition (O 2p → Fe 3d) centered at 3.2 eV⁵⁵ can effectively occur when being excited by short-wavelength light..." This explains the shift for Fe₂O₃ MCs, however, there should be a different bond behavior for Ti doped situation both in emission band and PL decay. If it is not the case, then Ti is not influencing the Fe₂O₃ structure, which supports the idea that Ti is mostly segregated at the interfaces of Fe₂O₃. This needs to be clarified by the authors. From my perspective the data

could all be explained with segregation.

Reply: Thanks to the reviewer's comments. According to the fact that the intensity of PL band II obtained for Ti-Fe₂O₃ MCs is weaker than that for pure Fe₂O₃ MCs, it seems that this emission is closely related to the oxygen vacancy. Meanwhile, the shortened lifetime for Ti-Fe₂O₃ MCs in comparison with pure Fe₂O₃ MCs suggests that the charge separation becomes more facile by Ti-doping. This is most probably due to charge separation by increased conductivity, not due to charge recombination, taking the superior PEC performance of Ti-Fe₂O₃ MCs into account. The recent results obtained by time-resolved microwave conductivity indicated that donor doping increases the effective mobility of hematite dramatically (Adv. Funct. Mater. 2019, 1901590) as discussed in the revised manuscript (page 23). To further understand the impact of metal doping on the PL behavior, a much more theoretical consideration will be performed in near future. In addition, rutile TiO₂ exhibits defect-mediated emission in the near-infrared region (around 830 nm) (J. Phys. Chem. C 2017, 121, 9011–9021). Actually, emission from rutile TiO₂ was not observed for Ti-Fe₂O₃ MCs. We added this result in the revised manuscript (page 22) as follow: "It should be noted that there is no emission from rutile TiO₂, which is known to exhibit defect-mediated emission in the near-infrared region (~830 nm),⁶¹ for Ti-Fe₂O₃ MCs."

13) EPR measurement confirmed the presence of Fe²⁺. The presence of high amount Fe²⁺ may act trapping electrons instead of favor the electron hopping mechanism in hematite. How the presence of Fe²⁺ could positively affect the charge transport? In addition, several papers showed that the presence of Fe²⁺ in hematite provide inefficient surface for chemical reaction.

Reply: We found that abundant oxygen vacancy can form at the sintering interfaces in MC after thermal treatment according to the analysis of STEM-EELS, XPS, EPR, and steady-state UV-visible diffuse reflectance spectral measurements. Notably, the electron concentration of the Fe₂O₃ MC photoanode is almost 6.6 times higher than that of the Fe₂O₃ SC photoanode. Therefore, the abundant bulk V_O can actually increase the bulk carrier density and thus effectively improve the electrical conductivity. To analyze the role of defects at the surfaces, PEIS measurements were carried out. It is obvious that the density of surface states (N_{SS}) in MC photoanodes are both much lower than that of Fe₂O₃ SC electrode, suggesting that smaller numbers of holes will be trapped in the surface states at semiconductor-electrolyte interface of MC electrodes as given in Fig. 4d. It has been mentioned that excess Fe²⁺ might enhance the recombination with holes and/or suppress the generation of holes (Fe⁴⁺) because total Fe concentration is constant (Sustainable Energy Fuels, 2019, 3, 1351-1364). Nevertheless, our results indicate that the presence of Fe²⁺ inside MCs would positively affect the charge transfer

owing to the enhanced conductivity.

The responses to the comments from Reviewer #3:

In this work, the authors reported the preparation of novel thick hematite films (~1500 nm) constructed by highly ordered and intimately attached hematite mesocrystals (MCs) for highly efficient PEC water oxidation. The study is somehow comprehensive and well organized. The authors attributed the enhanced PEC water oxidation to the high carrier density derived from oxygen vacancies. But the authors neglect the role of surface states presented in the hematite/electrolyte interfaces, which is very important for the charge transfer during PEC water splitting. Additionally, there are some technical issues need to be addressed. Thus, I recommend it to be major revised before published on “Nature Communications”. My detail suggestions are as follows:

The authors express sincere thanks to the reviewer for his/her valuable comments and constructive suggestions.

(1) The TEM images in Figure 1a and 1d reveal that discus-like structure is composed of some nanoparticles, indicating the whole discus-like structure is not a single crystalline. But the authors show a SAED with a single set of diffraction pattern, which is an indicative of single crystalline? The authors should indicate the accurate region where they got the SAED data!

Reply: Thanks to the reviewer’s kind comment and suggestion. The SAED was captured from the whole MC particle region. The result exhibited the single-crystalline diffraction spot, thus indicating all the nanoparticles inside of the MC particle have the same orientation. In the revised manuscript, we have clearly indicate the region for SAED capture. On page 5, “The corresponding selective area electron diffraction (SAED) pattern captured over the whole region of MC shows single-crystal-like spots (Fig. 1a, inset), indicating that the nanocrystal subunits are highly ordered and crystallographically aligned, which can be attributed to particle-to-particle interaction.”; On page 6, “The TEM images and corresponding SAED pattern obtained from the whole MC particle in Fig. 1d indicate that Ti-Fe₂O₃ MCs maintain their size, morphology, and crystal structure even after high temperature annealing at 700 °C (Fig. 1d and Supplementary Fig. 5a,b).”.

(2) The fusion interfaces with distorted lattice fringes in Figure 1d, meaning abundant defects, which could be dislocations, twins, vacancy etc. There is no directly TEM data can demonstrate

the fusion interfaces is derived from oxygen vacancies. Since the authors relate the observed fusion interfaces with the oxygen vacancies in the discussion part, in order to verify the presence of oxygen vacancies, the authors should use the relative EELS composition map to quantitative the oxygen elements content and use spherical aberration correcteto directly observe the oxygen atomic column. The author may refer to these references:

[1] Synergistic Effects in 3D Honeycomb-like Hematite Nanoflakes/Branched Polypyrrole Nanoleaves Heterostructures as High-Performance Negative Electrodes for Asymmetric Supercapacitors, *Nano Energy*, 2016, 22, 189-201.

[2] High-yield Synthesis and Optical Properties of g-C₃N₄, *Nanoscale*, 2015, 7, 12343-12350.

[3] Polarity assignment in ZnTe, GaAs, ZnO and GaN-AlN nanowires from direct dumbbell analysis, *Nano Letters*, 2012, 12, 2579-2586.

Reply: Thanks to the reviewer's suggestions. We have carried out the STEM-EELS experiment to clarify the tiny structure of the sample and added the content in the revised manuscript (pages 9-11) as follows: "HAADF-STEM with electron energy loss spectroscopy (EELS) measurements can give useful information about the local composition, structure, and chemical states of the samples.³²⁻³⁴ Fig. 2a shows the HAADF-STEM image of a typical Ti-Fe₂O₃ MC particle after being annealed at 700 °C in air. The corresponding EELS composition map is given in panel b. It is obvious that most of the Ti species are segregated to the outer surface and edges of the pores. To further analyze the distribution of the elements in the crystal, EEL spectra were acquired in different regions (Fig. 2a). As shown in Fig. 2c, only Ti-L_{2,3} and O-K signals were detected on the edge of the MC particle (Region 1 in Fig. 2a), suggesting the presence of TiO₂ overlayer. The thicknesses of TiO₂ layers are estimated from <1 to approximately 4 nm (Fig. 2b and Supplementary Fig. 11). It is noteworthy that TiO₂ (most probably rutile phase) grew on the surface of hematite as suggested by the high-resolution HAADF-STEM image, the corresponding fast Fourier transform (FFT) pattern (Supplementary Fig. 11), and the XRD peak attributed to rutile (110) (Supplementary Fig. 12). On the smooth surface (Region 2 in Fig. 2a), Ti-L_{2,3} signals are very weak, implying that a small portion of Ti⁴⁺ ions were doped into the crystal. A higher concentration of Ti⁴⁺ ions were observed near the edges of the pores (Region 3 in Fig. 2a), again indicating the formation of thin TiO₂ layers.

The synchrotron-based X-ray total scattering and pair distribution function (PDF) analyses were further performed to investigate the local structural changes in MCs (Supplementary Fig. 13). For Fe₂O₃ MCs, no apparent difference in Fe-O and Fe-Fe bond distances was observed between the samples before and after the annealing. This is probably due to the fact that the amounts of V_O in annealed MCs are still too small to be detected. Whereas, the Ti modification results in a significant intensity decrease and broadening of the peaks. After the annealing, the

spectral features are almost restored to those of pure hematite, indicating that most Ti ions are segmented from the bulk to the surface while leaving a trace as dopant.

To prove the formation of abundant interfacial V_O , high-resolution HAADF-STEM measurements combined with EELS maps were carried out at typical region with sintered interfaces (Figs. 2d, e). The valence states of iron were identified by Fe-L_{2,3} EEL spectra (Fig. 2f) and separately visualized in Fig. 2e using multiple linear least-square fit to the spectrum image data with the reference spectra extracted from the well-defined areas. Fe^{2+} ions (i.e., V_O)³⁵ are mostly distributed in the dark zone where the distorted lattice fringes are seen (Fig. 2d), while Fe^{3+} ions are distributed in the bright zone where clear lattice fringes are seen. The multivariate analysis³⁶ of spectrum imaging data was further performed to clarify the associations between the specific elements (Supplementary Fig. 14). The analytical results imply that Fe^{2+} species originate not only from V_O at the interface in MCs (Fig. 2e), but also from $Fe_{2-x}Ti_xO_3$ (e.g., ilmenite) on the surface. The latter is considered to be present at the interface between hematite and TiO_2 (Supplementary Fig. 11) and play a key role in the formation of the TiO_2 overlayer during the annealing.³⁷ Although the direct imaging of oxygen atomic column at the interface is still challenging at this time because of the stacking of primary nanocrystals in MC, we believe that the present results reasonably support our hypothesis.

Besides, all the suggested literature were cited in the revised manuscript.

(3) The grain sizes of $Ti-Fe_2O_3$ and Fe_2O_3 MCs are about several hundred nanometres, which is distinctly different from the value (30 nm) obtained from XRD. The authors should explain it!

Reply: Thanks to the reviewer's comment. As the $Ti-Fe_2O_3$ and Fe_2O_3 MCs with sizes of several hundred nanometers are both constructed by highly ordered and intimately attached nanoparticles with diameter of around 30 nm. XRD patterns can give information of crystal orientation and size of the primary nanoparticles inside MCs. According to the Scherrer equation, the size of nanoparticles inside MCs is approximately 30 nm, which is consistent with the diameter measured from TEM image. We revised the manuscript (page 8) as follows: "The grain sizes in $Ti-Fe_2O_3$ and Fe_2O_3 MCs calculated using Scherrer equation are ~30 nm, which is close to the sizes of the nanocrystal subunits in as-synthesized MCs verified from TEM images (Supplementary Figs. 2 and 7)."

(4) In Figure S1, except the discus-like structure, there is also some rugger-like structure; is it a residual or a mixture of discus-like structure and rugger-like structure?

Reply: Thanks for the reviewer's question. Actually, the rugger-like structure observed in Fig.

S1 is the side view of the discus-like structure. The rugger-like can also be seen from the cross-sectional SEM image of Ti-Fe₂O₃ MC electrode in Fig. 1h.

(5) In table S1, why the R_s reduces so much in the case of Ti-Fe₂O₃ MCs? Where is the data of C_{trap}? Where is the flatland data of these three electrodes and how does they change in these three electrodes? The authors should present all the important data, including flatland and C_{trap} from EIS and Mott-Schottky plots in the manuscript and SI!

Reply: Thanks to the reviewer's suggestions. In the revised manuscript, we have added all the suggested data in Supplementary Table 2 to clarify the difference.

(6) As show in Fig. 5C, there are always different defects presented at the surface of hematite, mediating the charge transfer at the hematite/electrolyte interfaces. The authors solely attributed the enhanced PEC performance to the enhanced electronic conductivity resulted from the improved oxygen vacancy but neglected the possible change of the surface states in hematite. Is there any change of the surface states in hematite upon Ti doping and nanostructuring? What is the role of the surface states in hematite during the PEC water splitting? The authors should refer to books from Juan Bisquert: Photoelectrochemical solar fuel production—from basic principles to advanced devices, [M]. 2016, Berlin: Springer and the following references:

[1] Water oxidation at hematite photoelectrodes: The role of surface states, *J. Am. Chem. Soc.*, 2012, 134, 4294-4302.

[3] Understanding the role of underlayers and overlayers in thin film hematite photoanodes, *Adv. Funct. Mater.* 2014, 24, 7681-7688.

[4] Electrochemical and photoelectrochemical investigation of water oxidation with hematite electrodes, *Energy Environ. Sci.*, 2012, 5, 7626-7636.

[5] Water oxidation on hematite photoelectrodes: insight into the nature of surface states through in situ spectroelectrochemistry, *J. Phys. Chem. C*, 2014, 118, 10393-10399.

[6] Enhanced water splitting efficiency through selective surface state removal, *J. Phys. Chem. Lett.*, 2014, 5, 1522-1526.

[7] What do you do, Titanium? Insight into the Role of Titanium Oxide as Water Oxidation Promoter in Hematite-based Photoanodes, *Energy & Environmental Science*, 2015, 8, 3242-3254.

[8] Determination of photoelectrochemical water oxidation intermediates on haematite electrode surfaces using operando infrared spectroscopy, *Nat. Chem.*, 2016, 8, 778-783.

[9] Enhanced Photoelectrochemical Water Splitting of Hematite Multilayer Nanowires

Photoanode with Tuning Surface State via Bottom-up Interfacial Engineering, *Energy & Environmental Science*, 2017, 10, 2124-2136.

The authors should systematically monitor the donor density and surface states evolution simultaneously via PEIS and EIS under dark. The authors should clearly clarify the contributions of enhanced electronic conductivity and changed surface states to the enhanced PEC water splitting, respectively!

Reply: According to the reviewer's suggestions, we have added the PEIS experimental results in the revised manuscript (pages 18-19) as follows: "The surface state (SS) is known to strongly influence charge transfer processes at the SEI.⁴⁷⁻⁵⁵ To systematically study the charge transfer in the MC electrodes, photoelectrochemical impedance spectroscopy (PEIS) measurements were performed at potentials ranging from 0.7 to 1.6 V vs. RHE (Supplementary Fig. 19). The density of mid-gap surface states (N_{SS}), which are considered to be related to OH^- and O-terminated hematite surfaces,⁵⁶ was calculated from C_{trap} according to Supplementary Eq. 4 and shown in Fig. 4c. The peaks of the energetic distribution of N_{SS} are observed near the formal potential for water oxidation (1.23 V vs. RHE), which decides an equilibration of trap-state energy and hole-accepting species at SEI.⁴⁹ The N_{SS} values of MC electrodes are both lower than that of the SC electrode; this suggests that surface states are reduced via oriented attachment of primary nanocrystals, possibly to reduce the overpotential and enhance the photovoltage.^{51,57,58} These results again support that the hierarchical construction of highly ordered MCs with same crystal orientation would effectively improve the PEC performance. In comparison with the Fe_2O_3 MC electrode, the slightly increased N_{SS} of Ti- Fe_2O_3 MC electrode might be due to the formation of TiO_2 overlayer on the surface. The correlation between N_{SS} , N_d , and current density at 1.23 V vs. RHE is demonstrated in Fig. 4d. The higher N_d , which originates from the abundant interfacial V_O and/or Ti incorporation, and lower N_{SS} of MCs would effectively generate higher photocurrents."

Along with the data in Supplementary Table 2, we have added the explanation "The fitting results according to the equivalent circuit model are summarized in Supplementary Table 2. The series resistance (R_s) at the interface between the FTO substrate and hematite layers represents a substantial reduction from ca. $100 \Omega \text{ cm}^2$ for the unmodified sample to ca. $40 \Omega \text{ cm}^2$ for the Ti- Fe_2O_3 MC sample. This result suggests the possibility that the formation of TiO_2 layers on the surface of hematite MCs effectively improves the electron transfer from hematite to the FTO substrate. The charge transfer resistance (R_{ct}) increased in the order of Ti- Fe_2O_3 MC ($367 \Omega \text{ cm}^2$) < Fe_2O_3 MC ($576 \Omega \text{ cm}^2$) < Fe_2O_3 SC ($2747 \Omega \text{ cm}^2$), indicating the superior charge mobility in the bulk of MC due to the increased carrier density by V_O and Ti modification, as

well as the highly ordered MC alignment. R_{trap} reflects the charge transfer property at the semiconductor/electrolyte interfaces (SEI). The MC samples exhibited lower R_{trap} values than that of the SC sample, indicating a higher charge transfer efficiency at SEI.” in the revised manuscript (pages 17-18).

Besides, all the suggested literatures were cited in the revised manuscript.

(7) In Figure 4. f, The proposed diffusion processes of charge carriers under different illumination modes should be Figure 4. e, The proposed diffusion processes of charge carriers under different illumination modes.

Reply: According to the reviewer’s comment, we have correct the legend in the revised manuscript.

(8) There seems to be no data reporting the detection or quantification of oxygen. As this is a paper about photoelectrochemical water oxidation (presumably to produce oxygen), that seems to be an omission. I admit that it is highly likely that oxygen is the main product, but as it stands an unkind referee could argue that the lack of this oxygen data undermines the conclusions of the paper. I therefore suggest that some oxygen detection data are included, and a Faradaic yield for oxygen quoted.

Reply: We appreciate for the reviewer’s kind suggestion. In the revised manuscript, we added the result of the PEC system using Ti-Fe₂O₃ MC photoanode (Fig. 3c). As shown in Fig. 2e, gases were evolved from Ti-Fe₂O₃ MC photoanode and Pt counter electrode at 1.23 V vs. RHE with an irradiation area of 0.20 cm². Both H₂ and O₂ were linearly produced over 3 hours with the stoichiometric ratio (2:1). The Faradaic efficiencies (FEs) for both H₂ and O₂ were approximately 97%.

Once again we would like to express our heartfelt thanks to the reviewers. Their fruitful comments and constructive suggestions helped us to improve the quality of the paper significantly.

Reviewers' comments:

Reviewer #1 (Remarks to the Author):

This article is about photoelectrochemical performance of mesocrystalline MC hematite. The authors study thick films of about 1500 nm and find that photocurrent is higher under back illumination. The MC are composed with nanocrystal subunits of about 30 nm are compared with SC hematite which are composed with nanocrystal of 10 nm. They find that the photocurrent is higher for the MC sample and it can be also improved by Ti modification. They characterized the samples by numerous experimental methods (TEM, XPS, PL...) and try to understand why the MC samples has the better photocurrent. The authors claim that the improvement is due to interfacial oxygen vacancies which are formed in the MC sample during heating at 700 ° in air.

The second version of the paper with major modifications is more convincing than the first version, however some points are still unclear.

1- according to the figure 2, and fig 10 in sup., it seems that Fe²⁺ is present only when Ti percentage is high (near the surface where a layer of TiO₂ is formed or inside the sample where the Ti amount is high). Indeed we clearly see on figure 10 supp. that the valence of iron shift from Fe²⁺ near the surface toward Fe³⁺ at 20 nm from the surface. Moreover for the Fe₂O₃ MC without titanium there is no experimental proof of the presence of Fe²⁺. Therefore from my point of view the presence of Fe²⁺ is linked to inhomogeneous segregation of titanium in sample and locally change in valence of iron. Indeed, as explained in my first report no oxygen vacancies can be introduced by air annealing in hematite.

2- As In point in the first review, XPS is sensible to only the first nm near the surface and therefore cannot detect oxygen vacancies inside crystallite of 30 nm! Therefore this technique cannot see interfacial oxygen vacancies. O1s XPS depth profile (if available) should be more useful.

3- concerning EIS experiment: I don't understand how the TiO₂ layer on the surface can influence the value of R_s (electron transfert between hematite and FTO). The TiO₂ layer situated on the surface should better influence the transfert between sample and electrolyte.

4-line 394 the authors write "the density of mid-gap surface state (N_{ss}) which are considered to be related to OH-....". It means that N_{ss} increases with OH- amount. This point implies that the amount of OH- is higher on Fe₂O₃-SC (see figure 4c), however it is in contraction with XPS measurements where where O_v and O_c (attributed to OH-) is lower on Fe₂O₃ SC (see fig 2c).

5-minor points:

-line 413 : fig 5a.

-Figure 5d, X label: thickness (nm)

As a conclusion I think that the interpretation of experimental data are still confusing. The arguments of the authors are not convincing. Therefore I do not recommend publication in nature com journal.

Reviewer #2 (Remarks to the Author):

I have read the revised manuscript carefully. The author had made corresponding adjustments to the places was pointed out. I recommend the paper for publication on nature communication.

Reviewer #3 (Remarks to the Author):

The authors addressed most of the questions. In principle, the present manuscript is suitable to be published on Nature Communications. But I would like to remind the authors to answer two minor questions before the final publish since these questions may raise the concerns from the readers of Nature Communication in the near further. My detail suggestions are as follows:

(1) In Supplementary Fig. 11, the detail crystalline zone axis of hematite and TiO_2 should be marked in the FFT images. There are more than three sets of diffraction spots in the FFT images, the author should indicate the detail crystalline planes in the red and blue cycles. In order to clearly show the interface of Fe_2O_3 and TiO_2 , the author may try to use IFFT images to get the phase map of TiO_2 and Fe_2O_3 .

(2) The EELS map in Figure 2 shows the segregation of Ti element at the edge of Ti- Fe_2O_3 MC nanoparticles, while the EDX map in Figure 1 shows a homogenous Ti element distribution though Ti- Fe_2O_3 MC nanoparticles. These data conflict with each other. Which is more believable? The author should clarify the difference.

Responses to the comments from Reviewer #1:

This article is about photoelectrochemical performance of mesocrystalline MC hematite. The authors study thick films of about 1500 nm and find that photocurrent is higher under back illumination. The MC are composed with nanocrystal subunits of about 30 nm are compared with SC hematite which are composed with nanocrystal of 10 nm. They find that the photocurrent is higher for the MC sample and it can be also improved by Ti modification. They characterized the samples by numerous experimental methods (TEM, XPS, PL...) and try to understand why the MC samples has the better photocurrent. The authors claim that the improvement is due to interfacial oxygen vacancies which are formed in the MC sample during heating at 700 ° in air.

The second version of the paper with major modifications is more convincing than the first version, however some points are still unclear.

We express sincere thanks to the reviewer for his/her valuable comments and constructive suggestions.

1- according to the figure 2, and fig 10 in sup., it seems that Fe²⁺ is present only when Ti percentage is high (near the surface where a layer of TiO₂ is formed or inside the sample where the Ti amount is high). Indeed we clearly see on figure 10 supp. that the valence of iron shift from Fe²⁺ near the surface toward Fe³⁺ at 20 nm from the surface. Moreover for the Fe₂O₃ MC without titanium there is no experimental proof of the presence of Fe²⁺. Therefore from my point of view the presence of Fe²⁺ is linked to inhomogeneous segregation of titanium in sample and locally change in valence of iron. Indeed, as explained in my first report no oxygen vacancies can be introduced by air annealing in hematite.

Reply: We appreciate for the kind suggestion. To confirm the existence of oxygen vacancies in pure hematite mesocrystals (Fe₂O₃ MCs) annealed at 700 °C in air, STEM-EELS analysis was carried out. The results show that abundant Fe²⁺ formed in the bulk of the mesocrystal. Furthermore, O 1s XPS depth profile also showed that the concentration of species related to oxygen vacancies remains unchanged from the surface to the bulk region (at ca. 20 nm depth) for both Fe₂O₃ and Ti-Fe₂O₃ MCs. Therefore, we can conclude that the formation of oxygen vacancies (i.e., Fe²⁺) inside the MCs was not just induced by the segregation of Ti to the surface, but mainly produced by sintering of interfaces of highly ordered and intimately attached nanoparticles inside the MC. Such an intimate interface would serve as oxygen-deficient environment which can contribute to the easier formation of oxygen vacancy even in air condition.

In addition to the replacement of figures (Fig. 2a-e and Supplementary Fig. 5), we revised the manuscript as “The properties of semiconductors are closely related to intrinsic lattice defects as well as their structure and composition. To analyze the defects in MCs, STEM-EELS analysis was carried out. Figs. 2a,b are the HAADF-STEM images of a typical Fe₂O₃ MC. As demonstrated in Figs. 2c and d, three components (1, 2, and 3) are clearly separated by multivariate analysis³⁵ of the spectrum image data obtained in the area of Fig. 2b. The components 1 and 2 can be assigned to oxides containing Fe³⁺ and Fe²⁺, respectively, form characteristic Fe-L_{2,3} spectra (Fig. 2e). These components are considered to be mostly located inside the MC from their high background intensity (Fig. 2d). Importantly, Fe²⁺ is distributed inside rather than the edge and pores without spatial overlap with Fe³⁺ (Fig. 2c). These features reflect the grain boundaries in the depth direction, although the interfaces between nanoparticles inside the MC itself almost disappeared due to sintering. In other words, V_O are likely to be formed in the regions where fusion of neighbored nanoparticles occurred during sintering. The component 3 probably originates from surface Fe³⁺ by taking into consideration its low background intensity and the L₃ peak slightly shifted to the lower energy side compared with those of the standard Fe³⁺ (Fig. 2e). Qualitatively similar results are obtained for different MCs (Supplementary Fig. 14).” on page 10, and “To further prove the presence of abundant V_O inside the MCs, O 1s XPS depth profiles were measured (Supplementary Fig. 18). It is clear that the concentration of V_O remains unchanged from the surface to the bulk region (at approximately 20 nm depth) for both Ti-Fe₂O₃ MC and Fe₂O₃ MC samples.” on page 13 of the revised manuscript.

2- As In point in the first review, XPS is sensible to only the first nm near the surface and therefore cannot detect oxygen vacancies inside crystallite of 30 nm! Therefore this technique cannot see interfacial oxygen vacancies. O1s XPS depth profile (if available) should be more useful.

Reply: Thank you for the reviewer’s suggestion. As already answered to the above comment, we have measured O 1s XPS depth profile and the results show that the concentration of oxygen vacancies remains unchanged from the surface to approximately 20 nm depth. This result again confirms the presence of oxygen vacancies inside the MC. We revised the manuscript as “To further prove the presence of abundant V_O inside the MCs, O 1s XPS depth profiles were measured (Supplementary Fig. 18). It is clear that the concentration of V_O remains unchanged from the surface to the bulk region (at approximately 20 nm depth) for both Ti-Fe₂O₃ MC and Fe₂O₃ MC samples” on page 13.

3- concerning EIS experiment: I don't understand how the TiO₂ layer on the surface can influence the value of R_s (electron transfer between hematite and FTO). The TiO₂ layer situated on the surface should better influence the transfer between sample and electrolyte.

Reply: In addition to the benefit of surface passivation, the formation of thin TiO₂ overlayer (<1-4 nm) between MCs and FTO would be considered to enhance electron tunneling at the interfaces as proposed for various oxide underlayers (e.g., TiO₂, Nb₂O₅ (*Adv. Mater.* 2012, 24, 2699), and Ga₂O₃ (*Far. Discuss.* 2012, 155, 223-232)). We mentioned this point in the revised manuscript (page 19) with the reference (ref. 47). In addition, due to the high temperature annealing, the interfacial properties (FTO/TiO₂/hematite) could be improved and significantly contribute to the charge transport and collection (*J. Mater. Chem. A*, 2015, 3, 5007-5013).

4-line 394 the authors write "the density of mid-gap surface state (N_{ss}) which are considered to be related to OH-...". It means that N_{ss} increases with OH- amount. This point implies that the amount of OH- is higher on Fe₂O₃-SC (see figure 4c), however it is in contradiction with XPS measurements where where O_v and O_c (attributed to OH-) is lower on Fe₂O₃ SC (see fig 2c).

Reply: According to the literature (*Angew. Chem. Int. Ed.* 2014, 53, 13404-13408), the surface state related to the OH-terminated surface is associated with occupied surface states extending over a broad range of energies in the bandgap, while the O-terminated surface is characterized by occupied surface states close to the top of the valence band. The OH-terminated surface is of particular interest in the context of (photo)electrochemistry because hematite undergoes spontaneous surface hydroxylation when exposed to aqueous electrolytes. Such surface states are related to the intermediate states formed via hole transfer to surface-coordinated water and concomitant deprotonation step during the water oxidation by metal oxide electrodes (*J. Am. Chem. Soc.* 2012, 134, 4294-4302). Considering that XPS analysis was not carried out in electrolyte, it might be reasonable that the concentrations of detected O_v and O_c have no direct correlation with the OH-terminated surface state.

5-minor points:

-line 413 : fig 5a.

-Figure 5d, X label: thickness (nm)

Reply: We have corrected the points in the revised manuscript. Thank you very much for the suggestions.

Responses to the comments from Reviewer #2:

I have read the revised manuscript carefully. The author had made corresponding adjustments to the places was pointed out. I recommend the paper for publication on nature communication.

Thank you very much for recommendation for publication. Again, we express sincere thanks to the reviewer for his or her valuable comments and constructive suggestions.

Responses to the comments from Reviewer #3:

The authors addressed most of the questions. In principle, the present manuscript is suitable to be published on Nature Communications. But I would like to remind the authors to answer two minor questions before the final publish since these questions may raise the concerns from the readers of Nature Communication in the near further. My detail suggestions are as follows:

Thank you very much for recommendation for publication. Again, we express sincere thanks to you for your valuable comments and constructive suggestions.

(1) In Supplementary Fig. 11, the detail crystalline zone axis of hematite and TiO₂ should be marked in the FFT images. There are more than three sets of diffraction spots in the FFT images, the author should indicate the detail crystalline planes in the red and blue cycles. In order to clearly show the interface of Fe₂O₃ and TiO₂, the author may try to use IFFT images to get the phase map of TiO₂ and Fe₂O₃.

Reply: According to the reviewer's suggestion. The diffraction spots in the FFT images were marked with crystalline planes. And IFFT images were added to compare the different phases of hematite and rutile (Supplementary Fig. 7).

(2) The EELS map in Figure 2 shows the segregation of Ti element at the edge of Ti-Fe₂O₃ MC nanoparticles, while the EDX map in Figure 1 shows a homogenous Ti element distribution though Ti-Fe₂O₃ MC nanoparticles. These data conflict with each other. Which is more believable? The author should clarify the difference.

Reply: EDX map has a lower resolution and mostly analyze the surface of a material, while chemical state mapping by STEM-EELS with multivariate analysis has a much higher

resolution to analyze the composition of a material in both surface and bulk. Therefore, to avoid misunderstanding, we remove the result of EDX-mapping of Ti-Fe₂O₃ MC.

REVIEWERS' COMMENTS:

Reviewer #1 (Remarks to the Author):

I have read carefully the third version of the manuscript. The authors made adjustments and answered most of the open questions, and now the arguments are convincing. Therefore, the present version is suitable for publication in Nature Communications.

One minor point: lines 257, 308, 309 and 812 : valence

Responses to the comments from Reviewer #1:

I have read carefully the third version of the manuscript. The authors made adjustments and answered most of the open questions, and now the arguments are convincing. Therefore, the present version is suitable for publication in Nature Communications.

Reply: Thank you very much for recommendation for publication. Again, we express sincere thanks to the reviewer for his or her valuable comments and constructive suggestions.

One minor point: lines 257, 308, 309 and 812 : valence

Reply: We have corrected the points in the revised manuscript. Thank you very much for the suggestions.